# Robust Filter Attention: Self-Attention as Precision-Weighted State Estimation

Peter Racioppo [1]

## Abstract

We introduce Robust Filter Attention (RFA), a formulation of self-attention as a robust state estimator. Each token is treated as a noisy observation of a latent trajectory governed by a linear stochastic differential equation (SDE), and attention weights are determined by consistency under this model rather than static feature similarity. Under isotropic noise and decay assumptions, RFA matches the computational complexity of standard attention. On language modeling benchmarks, RFA achieves lower perplexity than RoPE within the training window while remaining stable under zero-shot extrapolation to longer contexts. The framework also provides a dynamical interpretation of standard positional mechanisms, connecting rotational embeddings and recency biases to transport and uncertainty propagation induced by stochastic dynamics.

## 1. Introduction

Self-attention has become the dominant paradigm for sequence modeling due to its parallelism and scalability (Vaswani et al., 2017). Unlike recurrent architectures (Elman, 1990), however, it does not explicitly propagate latent states through shared temporal dynamics — each token attends independently to all others, with no constraint that states evolve consistently over time. Temporal structure is therefore encoded through positional mechanisms rather than state evolution.

We introduce Robust Filter Attention (RFA), which grounds such structure in an explicit stochastic dynamical prior: past tokens are propagated to the query position under learned linear dynamics, and attention weights are determined by their predicted reliability under this model, coupling transport and uncertainty propagation.

In RFA, each token is modeled as a noisy observation of a latent trajectory. The query token serves as a reference observation of the current state, while past keys are propagated to the query position under learned dynamics, each yielding a prediction of the current latent state with an associated precision determined by the Differential Lyapunov Equation (DLE). Attention weights are then computed from prediction errors measured under these precisions, so that tokens are weighted according to their consistency with the dynamical model. In this view, attention arises as precision-weighted state estimation under a stochastic dynamical prior. Unlike recursive filters such as the Kalman filter, RFA performs this estimation independently at each query position, yielding a parallel batch estimator compatible with standard attention-style computation.

Signal magnitude and reliability are governed separately: dynamical decay attenuates transported states, while DLE-derived precision tracks accumulated uncertainty. Depending on the learned balance between process and measurement uncertainty, different heads may specialize into distinct temporal filtering behaviors, ranging from short-range recency-biased filtering to stable long-range integration.

Common positional encodings can be understood as imposing implicit dynamical models on how information evolves across tokens. RoPE (Su et al., 2024) encodes transport through norm-preserving rotations but assigns no explicit reliability to tokens as a function of distance. ALiBi (Press et al., 2022) imposes a distance-based reliability penalty but carries no corresponding transport operator. In RFA, transport and reliability are not independent design choices but consequences of a shared dynamical model. RFA under noiseless, decay-free dynamics recovers RoPE, while pure diffusion with no decay yields a logarithmic distance penalty analogous to ALiBi.

Finally, we introduce Spectrally-Coupled RFA (SC-RFA). Standard rotational encodings such as RoPE apply no decay, so high-frequency components persist indefinitely. As a result, tokens far apart in the sequence can appear positionally similar to nearby ones, degrading long-context discrimination. SC-RFA addresses this by partitioning the frequency spectrum across attention heads and coupling each head's decay rate to its maximum frequency, so that high-frequency heads act as short-range filters while low-frequency heads

---

[1] Independent Researcher, Los Angeles, CA, USA. Correspondence to: Peter Racioppo <pcracioppo@gmail.com>.

*Proceedings of the $43^{rd}$ International Conference on Machine Learning*, Seoul, South Korea. PMLR 306, 2026. Copyright 2026 by the author(s).

behave as stable long-range integrators.

Our contributions are as follows:

**(i)** A derivation of self-attention as a tractable robust batch estimator for a linear SDE.

**(ii)** A scalable isotropic formulation with analytic uncertainty modeling at standard attention cost.

**(iii)** A recovery of RoPE and ALiBi as limiting cases, corresponding respectively to noiseless, zero-decay dynamics and Brownian diffusion.

**(iv)** A spectrally-coupled decay prior enabling multi-resolution temporal filtering across heads, with principled suppression of long-range phase interference.

## 2. Related Work

### 2.1. Probabilistic and Kernel Views of Attention

The Transformer architecture (Vaswani et al., 2017) computes attention scores via a scaled dot-product between queries and keys. While originally motivated by its efficiency and ability to capture long-range dependencies, a growing body of work has sought to interpret these weights as probabilities derived from latent statistical models.

Probabilistic Transformers (Gabbur et al., 2021) show that dot-product attention arises as a constrained limit of MAP inference in a Gaussian mixture model. The Bayesian Attention Mechanism (BAM) (Bianchessi et al., 2026) treats positional embeddings as explicit priors over token indices, while the Correlated Gaussian Process Transformer (CGPT) (Bui et al., 2024) interprets asymmetric projections through correlated Gaussian process inference. These approaches introduce probabilistic structure, but rely on static feature-space similarity or fixed prior covariances.

Other work has interpreted attention as a kernel regression estimator, modifying similarity geometry in feature space to improve robustness or variance properties (Tsai et al., 2019; Han et al., 2023; Nielsen et al., 2024; Liu et al., 2020; Goel & Bartlett, 2024). From this perspective, dot-product attention is a kernel smoother in which the kernel is determined by learned feature similarity, treating all tokens as equally reliable regardless of temporal lag. In contrast, RFA is not constructed by specifying or learning a similarity function. Instead, attention weights arise from a state estimation formulation under a shared dynamical model — the kernel over token pairs is a consequence of the underlying stochastic dynamics rather than the primary object of design.

### 2.2. Filtering, Continuous Dynamics, and SSMs

Continuous-time sequence models often parameterize latent dynamics using Neural Ordinary Differential Equations

(Neural ODEs) (Chen et al., 2018) and their stochastic extensions, Neural SDEs (Li et al., 2020; Shen & Cheng, 2025), which learn drift and diffusion functions from data. Several architectures integrate attention with continuous dynamics to handle irregular sampling or time-dependent relevance, including Continuous-Time Attention (Chien & Chen, 2021), Attentive Neural Processes (Kim et al., 2019), and ACE-NODE (Jhin et al., 2021). Self-Modulating Attention (SMA) (Chen et al., 2021) adjusts attention weights as a function of temporal distance.

Other work integrates neural networks with classical filtering frameworks by learning components of the Kalman filter, such as gains, noise models, or update rules (Jahanshahi & Zhu, 2026; Revach et al., 2022; Liu et al., 2023; Cohen & Klein, 2025; Shen et al., 2025). In contrast, RFA assumes a structured linear SDE whose DLE admits a closed-form solution, enabling parallel precision-weighted aggregation without learning Kalman gains or full covariance updates.

State space models (SSMs) provide another approach to sequence modeling by assuming linear time-invariant (LTI) dynamics and converting recurrence into convolution. Frameworks such as HiPPO (Gu et al., 2020) and S4 (Gu et al., 2022) achieve efficiency by restricting the dynamics to structured forms (e.g., diagonalizable or diagonal-plus-low-rank), reducing the cost of state updates from $\mathcal{O}(d^2)$ to $\mathcal{O}(d)$ and enabling fast convolutional implementations. Recent work shows that causal linear attention can be viewed as a special case of LTI convolution (Dao & Gu, 2024).

Concurrent with our work, Kalman Linear Attention (KLA) (Shaj et al., 2026) develops a probabilistic filtering interpretation for linear attention and state-space sequence models using parallelized information-form Kalman updates. Whereas KLA and recurrent SSMs propagate a compressed hidden state through recursive updates, RFA instead formulates self-attention as a parallel precision-weighted batch estimation problem, where pairwise uncertainties are analytically derived from stochastic dynamics rather than recursively propagated through posterior state updates, while preserving explicit content-based token interactions through attention.

### 2.3. Positional Encodings and Complex Geometry

Modeling relative temporal structure in Transformers has been approached with several geometric methods. RoPE (Su et al., 2024) encodes relative position through deterministic complex rotations of queries and keys, but introduces no explicit notion of decay or uncertainty. ALiBi (Press et al., 2022) applies a linear distance-based bias to attention logits, improving length extrapolation by suppressing distant interactions. xPos (Sun et al., 2023b;a) generalizes RoPE by combining rotations with dimension-wise decay to stabilize long-range behavior.

These methods impose useful geometric or monotonic structure, but are not derived from an explicit model of latent state evolution and measurement. As a result, transport and reliability are treated as separate design choices. Each method can be interpreted as corresponding to an implicit dynamical assumption. RoPE implements deterministic phase transport without uncertainty accumulation, corresponding to a noiseless LTI system with zero decay. ALiBi's linear distance penalty approximates RFA's DLE-derived bias under pure Brownian diffusion, giving it an implicit noise model but no transport operator. xPos introduces decay without deriving it from a covariance model, leaving transport and reliability decoupled.

RFA derives both the transport operator and the precision weighting from the same underlying SDE, so that these are coupled by construction rather than introduced independently. This also implies a rotate–aggregate–counter-rotate structure on the value stream, required for aggregation to correspond to fusion of state estimates in a shared temporal frame. While methods such as RoPER (Harik & Jayasiri, 2022) have applied value rotations previously, RFA derives this structure as a necessity of the dynamical model rather than a geometric heuristic.

Methods such as YaRN (Peng et al., 2024) address RoPE's extrapolation failure by compressing position indices at inference, mitigating out-of-distribution phase rotations post-hoc while introducing a train-inference mismatch. SC-RFA instead prevents the failure by coupling each head's decay rate to its maximum frequency during training, so that high-frequency modes attenuate before accumulating spurious phase matches. Selective RoPE (Movahedi et al., 2026) also combines rotation with decay via a spectral leakage analogy, but lacks a precision kernel, value rotation structure, and principled frequency-decay coupling — components that in RFA are necessary consequences of the dynamical model rather than design choices.

## 3. Methods

RFA models each token as a noisy observation of a latent trajectory evolving under a linear SDE. Keys are transported to the query position under the dynamics model, while the Differential Lyapunov Equation provides an analytic estimate of the uncertainty accumulated during transport. Attention weights are then determined by prediction consistency under this uncertainty model, yielding a robust precision-weighted estimator. Under isotropic noise and decay assumptions, the resulting attention kernel reduces to a scalar function of temporal lag, recovering standard $\mathcal{O}(N^2 d)$ attention complexity. Further details are provided in Appendix A and Appendix B.

### 3.1. Attention as Factorized State Estimation

**Generative model.** We model each token embedding $\boldsymbol{z}_i \in \mathbb{R}^d$ as a noisy observation of a latent state evolving under a linear time-invariant SDE:

$$
\begin{aligned}
d\boldsymbol{x}(t) &= \boldsymbol{A}\,\boldsymbol{x}(t)\,dt + \boldsymbol{G}\,d\boldsymbol{w}(t), \\
\boldsymbol{z}_i &= \boldsymbol{C}\boldsymbol{x}(t_i) + \boldsymbol{v}_i, \quad \boldsymbol{v}_i \sim \mathcal{N}(\boldsymbol{0}, \boldsymbol{R}),
\end{aligned}
\tag{1}
$$

where $\boldsymbol{w}(t)$ is a standard Wiener process, $\boldsymbol{v}_i$ is Gaussian measurement noise with covariance $\boldsymbol{R}$, and $\boldsymbol{Q} := \boldsymbol{G}\boldsymbol{G}^\top$ is the process noise covariance. We assume $\boldsymbol{C} = \boldsymbol{I}$ without loss of generality, since any invertible observation map may be absorbed into a change of basis; non-invertible maps correspond to partially observed systems, which we do not consider here.

This model is not intended to represent the true data-generating process. Rather, it defines a shared dynamical prior that specifies both how key embeddings are transported to the query position and how their reliability evolves. We adopt an LTI structure because it admits closed-form propagation of both state means and covariances.

**Estimation setup.** Given a query at position $i$, our goal is to estimate the latent state $\boldsymbol{x}(t_i)$ from past observations $\{\boldsymbol{z}_j\}_{j \leq i}$. This estimation problem is solved independently for each query position, yielding a parallel batch estimator rather than a recursive filter. At this stage, queries and keys are not separate representations, but roles assigned to the same embeddings. Each past embedding $\boldsymbol{z}_j$ serves as a key: it is propagated forward via the state transition matrix to form a prediction of the current latent state,

$$
\hat{\boldsymbol{z}}_{ij} = e^{\boldsymbol{A}\Delta t_{ij}} \boldsymbol{z}_j,
$$

where $\Delta t_{ij} = t_i - t_j$. The query embedding $\boldsymbol{z}_i$ serves as a reference observation against which these predictions are compared.

Under the SDE, the transported key is distributed as:

$$
\hat{\boldsymbol{z}}_{ij} \sim \mathcal{N}\big(\boldsymbol{x}(t_i),\ \hat{\boldsymbol{V}}_{ij}\big),
$$

where the covariance captures both accumulated process noise and the measurement noise of the source token:

$$
\hat{\boldsymbol{V}}_{ij} = \boldsymbol{V}(\Delta t_{ij}) + e^{\boldsymbol{A}\Delta t_{ij}} \boldsymbol{R}\, e^{\boldsymbol{A}^\top \Delta t_{ij}}.
$$

Here, $\boldsymbol{V}(\Delta t)$ is the solution of the Differential Lyapunov Equation (DLE):

$$
\dot{\boldsymbol{V}}(s) = \boldsymbol{A}\boldsymbol{V}(s) + \boldsymbol{V}(s)\boldsymbol{A}^\top + \boldsymbol{Q}, \qquad \boldsymbol{V}(0) = \boldsymbol{0}. \tag{2}
$$

**Query-key residual.** Because the query serves as a reference rather than a transported prediction, we assign it a

separate covariance $\boldsymbol{R}_\Gamma$, independent of $\boldsymbol{R}$, yielding a two-sided measurement model with distinct noise on the query and key sides.

The residual between query $\boldsymbol{z}_i$ and transported key $\hat{\boldsymbol{z}}_{ij}$ is:

$$\boldsymbol{r}_{ij} := \boldsymbol{z}_i - \hat{\boldsymbol{z}}_{ij}.$$

The residual covariance is the sum of the key-side and query-side uncertainties:

$$\boldsymbol{r}_{ij} \sim \mathcal{N}\Big(\boldsymbol{0}, \ \boldsymbol{\Sigma}_{ij}\Big), \qquad \boldsymbol{\Sigma}_{ij} = \underbrace{\hat{\boldsymbol{V}}_{ij}}_{\text{key-side}} + \underbrace{\boldsymbol{R}_\Gamma}_{\text{query-side}},$$

The positive definite query-side term $\boldsymbol{R}_\Gamma \succ 0$ ensures that the precision $\boldsymbol{P}_{ij} := \boldsymbol{\Sigma}_{ij}^{-1}$ remains bounded as $\Delta t \to 0$.

The similarity between query $\boldsymbol{z}_i$ and key $\boldsymbol{z}_j$ is measured by the squared Mahalanobis distance:

$$d_{ij}^2 = \boldsymbol{r}_{ij}^\top \boldsymbol{P}_{ij} \boldsymbol{r}_{ij}, \tag{3}$$

which replaces dot-product similarity with a consistency test under the precision prior given by the DLE.

**Latent state estimation.** The optimal estimator for $\boldsymbol{x}_i$ under the full joint model requires inverting a dense cross-token covariance matrix, which is not parallelizable. To obtain a tractable estimator, we adopt a conditional independence approximation, neglecting cross-token correlations induced by shared process noise while retaining the marginal covariance of each transported observation (Appendix A.3). We then estimate the latent state by minimizing a sum of squared Mahalanobis residuals:

$$\bar{\boldsymbol{z}}_i = \arg\min_{\boldsymbol{z}} \sum_{j \leq i} (\boldsymbol{z} - \hat{\boldsymbol{z}}_{ij})^\top \boldsymbol{P}_{ij} (\boldsymbol{z} - \hat{\boldsymbol{z}}_{ij}).$$

This yields the closed-form estimator:

$$\bar{\boldsymbol{z}}_i = \Big( \sum_{j \leq i} \boldsymbol{P}_{ij} \Big)^{-1} \sum_{j \leq i} \boldsymbol{P}_{ij} \, \hat{\boldsymbol{z}}_{ij}. \tag{4}$$

This approximation is exact when process noise is absent. In general, shared process noise induces cross-token correlations that the factorized likelihood ignores, causing correlated evidence to be overcounted. However, the marginal covariance of each transported observation is preserved, so the resulting precision weights remain individually consistent with the uncertainty predicted by the SDE. The approximation therefore sacrifices globally consistent inference in exchange for a tractable parallel attention-style estimator with structured uncertainty propagation.

**Robust reweighting.** The above estimator assumes residuals are well-described by the predicted covariance. To improve robustness under model mismatch, we instead use

a heavy-tailed likelihood over the Mahalanobis distance $d_{ij}^2$, yielding a robust penalty $\rho(d_{ij}^2)$, as in M-estimation.

Minimizing this loss yields data-dependent influence weights:

$$w_{ij} \propto \frac{\partial}{\partial d_{ij}^2} \rho(d_{ij}^2),$$

which re-weight the precisions:

$$\boldsymbol{P}_{ij} \to w_{ij} \, \boldsymbol{P}_{ij},$$

reducing the influence of tokens whose residuals are unexpectedly large under the predicted covariance.

Two standard choices are:

$$w_{ij} \propto \begin{cases} \exp\Big(-\frac{d_{ij}^2}{\nu}\Big) & \text{(exponential)} \\ \Big(1 + \frac{d_{ij}^2}{\nu}\Big)^{-\kappa} & \text{(power law)} \end{cases} \tag{5}$$

where $\nu$ governs the tail weight of the influence function. The exponential form recovers the standard dot-product attention structure, while the power-law form yields a heavier-tailed robust variant.

**Closed-form parallel computation.** To evaluate $\boldsymbol{P}_{ij}$ in closed form for all token pairs simultaneously, we require that the system matrices be simultaneously diagonalizable by some $\boldsymbol{S} \in \mathbb{C}^{d \times d}$: $\boldsymbol{A} = \boldsymbol{S}\boldsymbol{\Lambda}\boldsymbol{S}^{-1}$, $\boldsymbol{Q} = \boldsymbol{S}\boldsymbol{\Lambda}_Q\boldsymbol{S}^\dagger$, $\boldsymbol{R} = \boldsymbol{S}\boldsymbol{\Lambda}_R\boldsymbol{S}^\dagger$, $\boldsymbol{R}_\Gamma = \boldsymbol{S}\boldsymbol{\Lambda}_\Gamma\boldsymbol{S}^\dagger$. Under this assumption, the DLE decouples into independent scalar ODEs for each eigenmode $\boldsymbol{\Sigma}_{ij} = \boldsymbol{S}\boldsymbol{\Lambda}_{\Sigma,ij}\boldsymbol{S}^\dagger$, yielding a diagonal covariance in this basis. Letting $\boldsymbol{\lambda}_{(.)} = \text{diag}(\boldsymbol{\Lambda}_{(.)})$,

$$\boldsymbol{\lambda}_{\Sigma,ij} = \boldsymbol{\lambda}_Q \odot \frac{1 - e^{2\,\text{Re}(\boldsymbol{\lambda})\,\Delta t_{ij}}}{-2\,\text{Re}(\boldsymbol{\lambda})} + e^{2\,\text{Re}(\boldsymbol{\lambda})\,\Delta t_{ij}} \odot \boldsymbol{\lambda}_R + \boldsymbol{\lambda}_\Gamma.$$

The precision is obtained by diagonal inversion, $\boldsymbol{\lambda}_{P,ij} = \boldsymbol{1} \oslash \boldsymbol{\lambda}_{\Sigma,ij}$. Since the precision is diagonal in this basis, the Mahalanobis distance decomposes into independent scalar contributions per mode. Letting $\boldsymbol{z}_{s,i} = \boldsymbol{S}^{-1}\boldsymbol{z}_i$, the robust precision-weighted aggregation takes the form:

$$\bar{\boldsymbol{z}}_{s,i} = \sum_{j \leq i} \mathcal{A}_{ij} \odot \hat{\boldsymbol{z}}_{s,ij}, \qquad \hat{\boldsymbol{z}}_{s,ij} = e^{\boldsymbol{\Lambda}\Delta t_{ij}} \boldsymbol{z}_{s,j},$$

$$\mathcal{A}_{ij} := w_{ij}\boldsymbol{\lambda}_{P,ij} \oslash \Big( \sum_{j' \leq i} w_{ij'}\boldsymbol{\lambda}_{P,ij'} \Big),$$

When the eigenvalues of $\boldsymbol{A}$ are purely imaginary, $e^{\boldsymbol{\Lambda}\Delta t}$ reduces to element-wise rotations, as in RoPE.

### 3.2. Robust Filter Attention Mechanism

We instantiate the robust state estimator as a complex-valued attention layer by identifying the abstract diagonalization

matrices with learned linear projections. The input projections $\boldsymbol{W}_q, \boldsymbol{W}_k, \boldsymbol{W}_v \in \mathbb{C}^{d \times d}$ learn the transformation into the diagonalizing basis of the SDE's system matrices, absorbing the inverse diagonalizing matrix $\boldsymbol{S}^{-1}$, while the output matrix $\boldsymbol{W}_o$ absorbs $\boldsymbol{S}$, mapping the filtered estimates back to the original basis:

$$\boldsymbol{Q} = \boldsymbol{W}_q \boldsymbol{Z}, \quad \boldsymbol{K} = \boldsymbol{W}_k \boldsymbol{Z}, \quad \boldsymbol{V} = \boldsymbol{W}_v \boldsymbol{Z} \quad \in \mathbb{C}^{d \times N}.$$

To preserve the $\mathcal{O}(N^2 + Nd)$ memory complexity of standard attention, we impose isotropic decay and noise in the learned eigenbasis (per head):

$$\boldsymbol{\Lambda} = -\mu \boldsymbol{I} + i \boldsymbol{\Lambda}_\Omega, \ \ \boldsymbol{\Lambda}_Q = \sigma^2 \boldsymbol{I}, \ \ \boldsymbol{\Lambda}_R = \eta^2 \boldsymbol{I}, \ \ \boldsymbol{\Lambda}_\Gamma = \gamma^2 \boldsymbol{I},$$

where $\mu, \sigma^2, \eta^2, \gamma^2 \in \mathbb{R}^+$ and $\boldsymbol{\Lambda}_\Omega \in \mathbb{R}^{d \times d}$ is diagonal with $k$th diagonal entry $\omega_k$. This removes the ability to model dimension-dependent noise, but preserves the temporal dependence of uncertainty on lag. These definitions ensure marginally stable dynamics and positive semi-definite noise covariances.

Under isotropic decay and noise, each eigenmode follows independent exponentially decaying rotations with decay rate $\mu$ and angular frequency $\omega_k$. This yields simple element-wise rotation factors for forward/backward propagation, and a decay kernel that depends only on the time lag $\Delta t_{ij}$:

$$\tilde{\boldsymbol{\Phi}}^-[k, i] := e^{-i\omega_k t_i}, \ \tilde{\boldsymbol{\Phi}}^+[k, i] := e^{i\omega_k t_i}, \ \boldsymbol{E}[i, j] := e^{-\mu \Delta t_{ij}}.$$

We define rotated queries, keys, and values:

$$\tilde{\boldsymbol{Q}} := \tilde{\boldsymbol{\Phi}}^- \odot \boldsymbol{Q}, \quad \tilde{\boldsymbol{K}} := \tilde{\boldsymbol{\Phi}}^- \odot \boldsymbol{K}, \quad \tilde{\boldsymbol{V}} := \tilde{\boldsymbol{\Phi}}^- \odot \boldsymbol{V}.$$

The isotropic constraints cause the variance to become independent of the feature dimension:

$$\boldsymbol{\Sigma}_{\Delta t}[i, j] := \tilde{\sigma}^2 \left(1 - e^{-2\mu \Delta t_{ij}}\right) + \eta^2 e^{-2\mu \Delta t_{ij}} + \gamma^2. \quad (6)$$

Here, $\tilde{\sigma}^2 := \frac{\sigma^2}{2\mu}$, $\eta^2$, and $\gamma^2$ are learned scalar parameters (per head), corresponding respectively to steady-state process uncertainty, historical measurement noise (key-side), and uncertainty in the query observations; $\boldsymbol{\Sigma}_{\Delta t}$ denotes the scalar variance as a function of lag, distinct from the full covariance matrix.

Collecting terms,

$$\boldsymbol{\Sigma}_{\Delta t}[i, j] = \alpha e^{-2\mu \Delta t_{ij}} + \beta, \quad \alpha := \eta^2 - \tilde{\sigma}^2, \quad \beta := \gamma^2 + \tilde{\sigma}^2.$$

The behavior of the variance with temporal lag is governed by the sign of $\alpha$. When $\alpha < 0$, process noise dominates and precision decreases monotonically, yielding a *diffusive regime*. When $\alpha > 0$, measurement noise dominates and precision increases with lag, corresponding to an *integrative regime*. Because $\alpha$ and $\beta$ are determined by learned scalar parameters per head, different heads can specialize into distinct filtering behaviors.

The isotropic constraint allows the Mahalanobis distance for all pairs $(i, j)$ to be computed by element-wise multiplying a matrix of scalar precisions $\boldsymbol{P}_{\Delta t}[i, j] := 1/\boldsymbol{\Sigma}_{\Delta t}[i, j]$ by a matrix of squared residual norms $\|\boldsymbol{R}_{qk}[i, j]\|^2$:

$$\boldsymbol{D}^2[i, j] = \boldsymbol{P}_{\Delta t}[i, j] \cdot \left\|\boldsymbol{R}_{qk}[i, j]\right\|^2,$$

where the $ij$th residual is:

$$\boldsymbol{R}_{qk}[i, j] := \tilde{\boldsymbol{Q}}_i - \boldsymbol{E}[i, j] \cdot \tilde{\boldsymbol{K}}_j.$$

The squared residual norm decomposes into a query magnitude term, a decayed key magnitude term, and a cross-term containing the complex inner product:

$$\begin{aligned}
\left\|\boldsymbol{R}_{qk}[i, j]\right\|^2 &= \|\boldsymbol{Q}_i\|^2 + \boldsymbol{E}[i, j]^2 \cdot \|\boldsymbol{K}_j\|^2 \\
&\quad - 2\,\boldsymbol{E}[i, j] \cdot \mathrm{Re}\left(\tilde{\boldsymbol{Q}}_i^\dagger \tilde{\boldsymbol{K}}_j\right).
\end{aligned} \quad (7)$$

Choosing the power-law form for $w_{ij}$ in Eq. 5 yields the following attention logits:

$$\boldsymbol{L} = \log(\boldsymbol{P}_{\Delta t}) - \kappa \log\left(1 + \frac{1}{\nu} \boldsymbol{P}_{\Delta t} \odot \left\|\boldsymbol{R}_{qk}\right\|^2\right), \quad (8)$$

where setting $\kappa = \frac{\nu + d}{d}$, and $\nu = \nu_s d$ for $\nu_s \in \mathbb{R}^+$ yields a dimension-free logit equivalent to a dimension-normalized Student-$t$ log-likelihood.

The attention matrix is then $\hat{\boldsymbol{A}} = \boldsymbol{A} \odot \boldsymbol{E}$, where:

$$\boldsymbol{A} = \mathrm{Softmax}_j\left(\beta_s \boldsymbol{L} + \boldsymbol{M}_{\mathrm{causal}}\right),$$

where $\beta_s$ is an additional inverse temperature parameter and $\boldsymbol{M}_{\mathrm{causal}} \in \{0, -\infty\}^{N \times N}$ is a causal mask.

The filtered value estimate is computed by aggregating the rotated values and rotating the result back into the original value frame:

$$\bar{\boldsymbol{V}} = \tilde{\boldsymbol{\Phi}}^+ \odot \left(\tilde{\boldsymbol{V}}\,\hat{\boldsymbol{A}}^\top\right).$$

This rotate–aggregate–counter-rotate structure is required by the dynamical model: values from different time steps must be brought into a common temporal frame before aggregation, and the counter-rotation restores the output to the original frame.

The attention layer then computes an innovation step in the value basis,

$$\Delta \boldsymbol{V} = \bar{\boldsymbol{V}} - \boldsymbol{V},$$

which represents a correction from the current value toward the filtered estimate. This correction is projected back into the original basis and added to the residual stream:

$$\boldsymbol{Z}^+ = \boldsymbol{Z} + \boldsymbol{W}_o \Delta \boldsymbol{V}.$$

Under the isotropic constraint, RFA preserves the asymptotic complexity of standard attention. The dominant operation remains a single $\mathcal{O}(N^2 d)$ matrix multiplication to

compute the cross-term $\operatorname{Re}(\tilde{\boldsymbol{Q}}^{\dagger}\tilde{\boldsymbol{K}})$. The remaining components—the decay kernel $\boldsymbol{E}[i,j]$ and precision kernel $\boldsymbol{P}_{\Delta t}[i,j]$—are computed via elementwise operations with $\mathcal{O}(N^2)$ cost, and do not change the asymptotic complexity.

### 3.3. Real-valued implementation

Although RFA is formulated over $\mathbb{C}^d$, all operations reduce to standard real arithmetic, as detailed in Appendix D.1. The complex projections $\boldsymbol{W}_q, \boldsymbol{W}_k, \boldsymbol{W}_v \in \mathbb{C}^{d \times d}$ are implemented as real $d \times 2d$ matrices and $\boldsymbol{W}_o$ as a $2d \times d$ matrix, while complex rotations reduce to the standard RoPE operation on paired real and imaginary channels (Appendix D.1). RoPE corresponds to restricting the eigenvalues of $\boldsymbol{A}$ to the imaginary axis ($\mu = 0$), so that the complex exponential $e^{\boldsymbol{\Lambda}\Delta t}$ reduces to pure rotation with no decay.

The complete implementation of the Isotropic RFA mechanism is formalized in Algorithm 1 in Appendix D.3. [1]

### 3.4. Iterative refinement across layers

The robust M-estimator defines a fixed-point problem: the weights $w_{ij}$ depend on residuals computed relative to the unknown latent state, which must be approximated by the current estimate. A single attention layer performs one such reweighting step using the previous layer's output as the current state estimate, while residual connections implement a partial step toward each reweighted estimate.

Stacking layers therefore yields an iteratively reweighted least squares (IRLS)-like procedure, in which observations are progressively reweighted according to their consistency with the evolving estimate. This provides an interpretation of both depth and residual connections in RFA as successive refinement of the state estimate (Appendix C.3).

### 3.5. Filtering Behaviors

The attention weights in RFA are determined by two distinct mechanisms that play complementary roles. The additive bias $\boldsymbol{B}_{\Delta t} := \log(\boldsymbol{P}_{\Delta t})$ acts as a prior budget allocated to tokens at each lag, while the multiplicative gate $\boldsymbol{P}_{\Delta t}$ controls the selectivity of the attention — how sharply the model discriminates between consistent and inconsistent tokens at that lag. Both are determined by the learned noise parameters $(\tilde{\sigma}^2, \eta^2, \gamma^2)$ through the scalar variance $\boldsymbol{\Sigma}_{\Delta t} = \alpha e^{-2\mu\Delta t} + \beta$.

The sign of $\alpha = \eta^2 - \tilde{\sigma}^2$ defines a phase transition between two qualitatively distinct filtering behaviors:

**Diffusive Regime** ($\alpha < 0$). When process noise dominates ($\tilde{\sigma}^2 > \eta^2$), uncertainty accumulates monotonically with lag.

---

The precision $\boldsymbol{P}_{\Delta t}$ acts as a *closing gate*: selectivity is maximal near the diagonal and degrades as lag increases. The additive bias decays toward a floor of $-\log(\beta)$, implementing a forgetting prior that suppresses distant tokens. These heads implement a recency bias, analogous to the linear distance penalties used in ALiBi (Appendix B.4).

**Integrative Regime** ($\alpha > 0$). When measurement noise dominates ($\eta^2 > \tilde{\sigma}^2$), the precision acts as an *opening gate*: selectivity is low near the diagonal and increases with lag as the initial measurement noise on the transported key dissipates under the stable dynamics. The additive bias correspondingly starts low and curves upward, implementing a settling prior that delays commitment until the transported observation becomes reliable. These heads function as lag-selective denoising filters, suppressing transient noise to identify stable historical structure.

Because $\alpha$ is determined by learned scalars per head, different heads can self-organize into different regimes during training. Explicit functional forms for the bias and gate in each regime are derived in Appendix B.4.

### 3.6. Recovery of Standard Positional Encodings

RoPE corresponds to the special case in which dynamics are noiseless ($\sigma^2 = 0$, $\eta^2 = 0$), decay is absent ($\mu = 0$), and value rotation is omitted. Under these conditions, the state transition matrix reduces to a complex rotation and the precision prior is uniform.

If the queries and keys are normalized, the Mahalanobis distance reduces to a dot product between queries and keys. In the zero-decay, short-lag limit, the RFA additive bias reduces to:

$$\boldsymbol{B}_{\Delta t} \approx -\log(\eta^2 + \gamma^2) - \frac{\sigma^2}{\eta^2 + \gamma^2}\,\Delta t,$$

recovering a linear distance penalty with learned slope and intercept, consistent with the form of ALiBi's fixed linear bias (Appendix B.3.5).

### 3.7. Spectrally Coupled RFA (SC-RFA)

Standard positional mechanisms such as RoPE utilize a fixed frequency bank across all heads, allowing high-frequency oscillations to persist indefinitely. At long horizons, this leads to phase wrap-around: tokens separated by a full oscillation period produce similar phase configurations, making them difficult to distinguish based on position alone.

To address this, we introduce Spectrally Coupled RFA (SC-RFA), in which decay rates are coupled to frequencies across heads. We partition a global frequency bank $\Omega$ monotonically across heads, assigning each head $h$ a spectral band $[\omega_{h,\min}, \omega_{h,\max}]$, and couple each head's decay rate to its

maximum frequency:

$$\mu_h = b \cdot \omega_{h,\max},$$

where $b \in \mathbb{R}^+$ is a dimensionless damping coefficient (Appendix B.5). This enforces a fixed decay per oscillation cycle: over one period, the signal is attenuated by a factor of $e^{-b}$, directly controlling the trade-off between spectral resolution and long-range stability.

Each attention head induces a prior temporal response given by the product of a decay term and a precision term:

$$\boldsymbol{P}_{\Delta t} \cdot e^{-\mu \Delta t} \ \propto \ \frac{e^{-\mu \Delta t}}{\alpha e^{-2\mu \Delta t} + \beta},$$

where $\alpha = \eta^2 - \tilde{\sigma}^2$ and $\beta = \gamma^2 + \tilde{\sigma}^2$.

In the integrative regime ($\alpha > 0$), precision initially increases with lag as key-side measurement noise dissipates, competing with exponential decay. The product peaks at a characteristic lag:

$$\Delta t^* = \frac{1}{2\mu} \log\left(\frac{\alpha}{\beta}\right).$$

Since $\mu_h$ varies across heads, peak locations $\Delta t^*_h \propto 1/\mu_h$ span a range of temporal scales, forming a bank of lag-selective filters. For sufficiently small $\mu$, the peak may lie beyond the training context length, causing the integrative profile to appear as a monotonically increasing function over the training window and inducing an anti-recency bias that extrapolates poorly. Diffusive heads ($\alpha < 0$) provide a fallback by enforcing monotonic decay and stabilizing long-range behavior.

# 4. Experimental Evaluation and Ablations

We evaluate whether explicitly modeling uncertainty growth improves long-context stability while preserving short-range accuracy, comparing RFA against two widely used positional baselines derived from deterministic geometry (RoPE) (Su et al., 2024) and monotonic recency biasing (ALiBi) (Press et al., 2022), respectively. The per-head scalar parameters $(\tilde{\sigma}^2, \eta^2, \gamma^2, \nu_s, \beta_s)$ are learned entirely within the training window ($L = 512$), and evaluated under zero-shot extrapolation at longer context lengths ($L \in \{512, 1024, 2048, 4096\}$).

## 4.1. Experimental Setup

**Architecture.** All models use a 6-layer Transformer with $h = 8$ heads and embedding dimension $d = 256$. To ensure comparable model capacity, we apply identical $d \to 2d \to d$ projections in both RFA and the RoPE/ALiBi baselines, which represent the mapping between the real and complex domains (Appendix D.1). RFA introduces only

a small number of additional scalar parameters per head for noise and robustness, increasing total parameter count by approximately 0.02%. We employ a pre-norm architecture with an FFN expansion factor of 4. Models are trained for 15 epochs until convergence using Adam with a cosine learning rate schedule.

**Datasets.** We evaluate on WikiText-103, a large-scale word-level language modeling benchmark derived from Wikipedia articles and used to measure perplexity and long-context extrapolation (Merity et al., 2017), and on BabyLM-2025 (Strict), a curated English language modeling corpus used as a complementary benchmark under the same training and evaluation protocol (Charpentier et al., 2025).

**Ablations.** We compare against standard positional baselines: RoPE (B1) and ALiBi (B2), and include two geometry-only decay variants to isolate the effect of damping in rotational embeddings: Decayed RoPE (B3), which applies exponential decay with distance, as in RFA, and SC-RoPE (B4), which couples decay rates to head-wise frequency bands, as in SC-RFA. These baselines test whether decay and spectral coupling alone can explain extrapolation gains, without modeling uncertainty.

We evaluate RFA (M1) and two variants of SC-RFA: one optimized for in-window performance (M2), with $b = 0.05$, and one optimized for long-context extrapolation via increased damping (M3), with $b = 5.0$. We also include structural ablations relative to M2, designed to isolate the effect of its components when removed: the power-law robust weight $w_{ij}$, replacing it with an exponential weight (M2.1); the DLE-derived precision prior (M2.2); the multiplicative gating term $\boldsymbol{P}_{\Delta t}$ (M2.3); value rotations (M2.4); all rotations (M2.5); finally, we test a purely rotational, zero decay and zero noise variant, analogous to RoPER (M2.6).

Full architectural and ablation details are provided in Appendix E. Analysis of attention maps and noise parameters are provided in Appendix F.

## 4.2. Results on Wikitext-103

We evaluate extrapolation by measuring test perplexity on WikiText-103 at increasing context lengths, after training all models with a fixed context window of 512 tokens. Results are shown in Table 1.

RFA variants achieve both stronger local performance and improved extrapolation relative to RoPE. In particular, SC-RFA (M2) improves over RoPE by 0.94 PPL at $L = 512$ and reduces degradation at long horizons, reaching 37.19 PPL at $L = 4096$ compared to RoPE's 72.69. M2 (SC-RFA) outperforms M1 (RFA) across all context lengths.

Unlike RoPE, which degrades monotonically outside the training window, RFA exhibits a non-monotonic extrap-

*Table 1.* Long-context extrapolation on WikiText-103 (Test PPL). All models were trained with a fixed context window of 512 tokens.

| Model | L=512 | L=1024 | L=2048 | L=4096 |
|---|---|---|---|---|
| RoPE (B1) | 28.48 | 30.94 | 44.21 | 72.69 |
| ALiBi (B2) | 28.59 | 27.30 | 26.54 | **26.30** |
| Decayed RoPE (B3) | 28.45 | 30.45 | 40.03 | 64.54 |
| SC-RoPE (B4) | 28.44 | 30.49 | 41.06 | 60.08 |
| RFA (M1) | 28.01 | 27.58 | 29.99 | 38.46 |
| **SC-RFA (M2)** | **27.54** | 26.73 | 29.46 | 37.19 |
| **SC-RFA (M3)** | 27.91 | **26.68** | **26.37** | 28.16 |
| *Structural Ablations (Relative to M2)* | | | | |
| Exp. Weight (M2.1) | 27.98 | 27.16 | 28.95 | 33.51 |
| Flat Prior (M2.2) | 27.69 | 28.71 | 38.11 | 62.83 |
| No Mult. Gate (M2.3) | 27.65 | 29.01 | 39.18 | 57.30 |
| No Value Rot. (M2.4) | 30.24 | 92.08 | 187.29 | 463.29 |
| No Rotations (M2.5) | 28.58 | 27.25 | 26.61 | 26.83 |
| Pure Rotation (M2.6) | 27.97 | 35.59 | 69.39 | 131.29 |

*Table 2.* Sensitivity analysis of the damping coefficient $b$ in SC-RFA (M2), with RoPE (B1) and ALiBi (B2) as baselines. Results show Test PPL on WikiText-103 across increasing context lengths.

| Damping ($b$) | L=512 | L=1024 | L=2048 | L=4096 |
|---|---|---|---|---|
| RoPE (B1) | 28.48 | 30.94 | 44.21 | 72.69 |
| ALiBi (B2) | 28.59 | 27.30 | 26.54 | **26.30** |
| $5 \times 10^{-4}$ | 27.60 | 28.88 | 37.34 | 51.48 |
| $5 \times 10^{-3}$ | 27.60 | 28.71 | 35.35 | 43.90 |
| $5 \times 10^{-2}$ | **27.54** | 26.73 | 29.46 | 37.19 |
| $5 \times 10^{-1}$ | 27.61 | **26.38** | **26.37** | 29.72 |
| $5 \times 10^{0}$ | 27.91 | 26.68 | **26.37** | 28.16 |

olation profile: perplexity initially decreases beyond the training horizon before increasing at longer context lengths.

With higher damping coefficient, SC-RFA (M3) further improves long-context stability, achieving nearly flat perplexity at up to $L = 4096$ while maintaining competitive performance within the training window. This behavior emerges under a fixed training protocol without requiring length-dependent scaling rules or curriculum schedules.

Introducing decay into rotational embeddings (B3) and spectrally coupling decay across heads (B4) slows the long-range degradation of RoPE. However, both geometry-only variants underperform RFA across all context lengths, indicating that decay alone is insufficient without explicit uncertainty modeling. B4 does not improve substantially over B3, indicating that coupling decay to the frequency spectrum alone provides little additional benefit.

The exponential weight variant of SC-RFA (M2.1) underperforms the power-law robust weighting at the training horizon, but achieves lower perplexity at extreme extrapolation lengths. This is consistent with Gaussian likelihoods imposing stronger quadratic penalties on residuals, which suppress extreme deviations more aggressively but reduce sensitivity to small errors when uncertainty is low.

Removing the DLE-derived precision prior (M2.2) leads to degradation at long horizons, with perplexity increasing to 62.83 at $L = 4096$, demonstrating that explicit uncertainty propagation via the DLE is necessary for stable long-context behavior. Removing the precision term $P_{\Delta t}$ from the Mahalanobis distance (M2.3) causes degradation within the training window and worsens extrapolation, indicating that both the additive and multiplicative precision terms contribute to stability.

Eliminating value-space rotation and counter-rotation (M2.4) causes severe degradation at long context, reaching 463.29 PPL at $L = 4096$. This is consistent with aggregation no longer corresponding to fusion of latent state estimates in a shared temporal frame. Removing all rotations (M2.5) degrades short-context performance but yields strong long-range stability. In this setting, RFA reduces to a purely distance-dependent bias with logarithmic scaling, closely matching the behavior and performance of ALiBi.

In the zero-noise, zero-decay, pure rotational setting (M2.6), perplexity increases sharply with context length. Despite outperforming RoPE within the training window, value rotations without decay and uncertainty modeling accelerate long-range degradation rather than preventing it.

Compared to ALiBi, SC-RFA achieves lower perplexity at the training length ($L = 512$) and at moderate extrapolation ($L = 1024$), suggesting improved utilization of fine-grained temporal structure when uncertainty remains low. At longer horizons, ALiBi attains lower perplexity by enforcing strict locality, effectively suppressing long-range interactions, while SC-RFA continues to integrate distant context with attenuated but nonzero precision. This reflects a trade-off between aggressive locality and uncertainty-weighted long-range integration.

Table 2 shows that this tradeoff is continuously controlled in SC-RFA by the damping coefficient $b$. Smaller values of $b$ yield slower decay, improving short-context performance but leading to faster degradation as context increases. Larger values of $b$ produce stronger attenuation and more stable long-range behavior at the cost of reduced short-range performance. Notably, for sufficiently strong damping ($b \geq 0.5$), SC-RFA outperforms ALiBi at intermediate horizons ($L = 2048$), with ALiBi retaining an advantage only at the largest tested context length.

### 4.3. Results on BabyLM-2025

We use the same architectures, hyperparameters, and training protocol as on WikiText-103. On BabyLM-2025, where

*Table 3.* Long-context extrapolation on BabyLM-2025 (Test PPL). All models were trained with a fixed context window of 512 tokens.

| Model | L=512 | L=1024 | L=2048 | L=4096 |
|---|---|---|---|---|
| RoPE (B1) | 17.70 | 18.78 | 23.33 | 33.29 |
| ALiBi (B2) | 17.70 | 17.20 | **17.06** | **17.51** |
| RFA (M1) | 17.51 | 17.71 | 20.61 | 31.04 |
| **SC-RFA (M2)** | **17.36** | **16.99** | 18.33 | 22.25 |
| **SC-RFA (M3)** | 17.51 | 17.07 | 17.26 | 18.74 |

language modeling performance is more strongly dominated by short-range context, differences between positional mechanisms are smaller at short context lengths. The RFA variants outperform RoPE at all evaluated context lengths and outperform ALiBi within the training window. SC-RFA also achieves lower perplexity than ALiBi at intermediate context ($L = 1024$), while ALiBi remains strongest at the longest horizons due to its strict recency bias. These results mirror the trade-off observed on WikiText-103: precision weighting improves robustness over purely rotational embeddings while retaining stronger short- and mid-range performance than aggressively local recency biases.

### 4.4. Learning Dynamics and Head Specialization

RFA variants achieve lower validation perplexity earlier in training than RoPE and ALiBi, indicating that the SDE-based prior provides an effective inductive bias for latent state estimation (Appendix F.1). Analysis of learned noise, decay, and robustness parameters reveals systematic specialization across heads into distinct uncertainty and selectivity regimes (Appendix F.2).

Attention map visualizations at long context lengths reveal structured multi-scale behavior in RFA. Compared to RoPE, attention maps exhibit reduced checkerboard interference from high-frequency oscillations and more coherent periodic structure, reflecting aggregation in a dynamically consistent frame. Some heads exhibit an integrative regime, in which attention is initially suppressed near the diagonal and peaks at a characteristic lag, indicating delayed aggregation until the latent state estimate stabilizes (Appendix F.3).

SC-RFA sharpens and organizes this structure through spectral coupling, inducing a clear ordering of temporal specialization across heads. High-decay heads concentrate attention near the diagonal, while lower-decay heads shift their mass toward progressively longer temporal offsets, producing distinct lag bands. This results in fewer and sharper periodic bands and a clearer separation between local and long-range interactions. Together, these patterns support the interpretation of RFA as learning a structured, uncertainty-aware temporal filter bank rather than relying solely on geometric positional bias.

## 5. Conclusion

We reformulate self-attention as a tractable approximation to state estimation under a linear stochastic dynamical model, yielding Robust Filter Attention (RFA). In this formulation, attention weights reflect uncertainty-aware agreement between dynamically transported representations rather than static feature similarity, while preserving the computational structure of standard attention. RFA recovers existing positional mechanisms as limiting cases and improves temporal consistency and long-context behavior.

Tractability requires four structural constraints — a conditional independence approximation, linear time-invariant dynamics, simultaneous diagonalizability, and isotropic noise and decay per head. A central challenge for future work is relaxing these constraints while retaining tractable covariance propagation and parallel computation.

Other important directions include understanding how these dynamical and uncertainty-based mechanisms interact with architectural components such as normalization and depth, as well as how similar uncertainty-aware formulations can be incorporated into other sequence modeling architectures.

## Impact Statement

This work introduces an uncertainty-aware formulation of self-attention. We do not identify any ethical concerns beyond those generally associated with advances in machine learning methodology.

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

# Appendix Table of Contents

## A. Attention from a Factorized State Estimation Formulation

Here, we derive the RFA mechanism introduced in Section 3 in greater detail.

We briefly recall the generative model from Section 3.1. The latent state evolves under a linear SDE,

$$d\boldsymbol{x}(t) = \boldsymbol{A}\boldsymbol{x}(t)\,dt + \boldsymbol{G}\,d\boldsymbol{w}(t), \qquad \boldsymbol{z}_i = \boldsymbol{x}(t_i) + \boldsymbol{v}_i,$$

where $\boldsymbol{w}(t)$ is a standard Wiener process, $\boldsymbol{v}_i$ is Gaussian measurement noise with covariance $\boldsymbol{R}$, and $\boldsymbol{Q} := \boldsymbol{G}\boldsymbol{G}^\top$ is the process noise covariance.

Each past token $\boldsymbol{z}_j$ is transported to time $t_i$ via:

$$\hat{\boldsymbol{z}}_{ij} = e^{\boldsymbol{A}\Delta t_{ij}}\boldsymbol{z}_j, \qquad \Delta t_{ij} = t_i - t_j.$$

Under the SDE, this transported observation satisfies:

$$\hat{\boldsymbol{z}}_{ij} \mid \boldsymbol{x}(t_i) \sim \mathcal{N}\big(\boldsymbol{x}(t_i), \hat{\boldsymbol{V}}_{ij}\big), \qquad \hat{\boldsymbol{V}}_{ij} = \boldsymbol{V}(\Delta t_{ij}) + e^{\boldsymbol{A}\Delta t_{ij}}\boldsymbol{R}\,e^{\boldsymbol{A}^\top \Delta t_{ij}},$$

where the accumulated process noise $\boldsymbol{V}(\Delta t_{ij})$ is the solution of the DLE (Section A.1). Figure 1 illustrates propagation through a stable LTI in 2D.

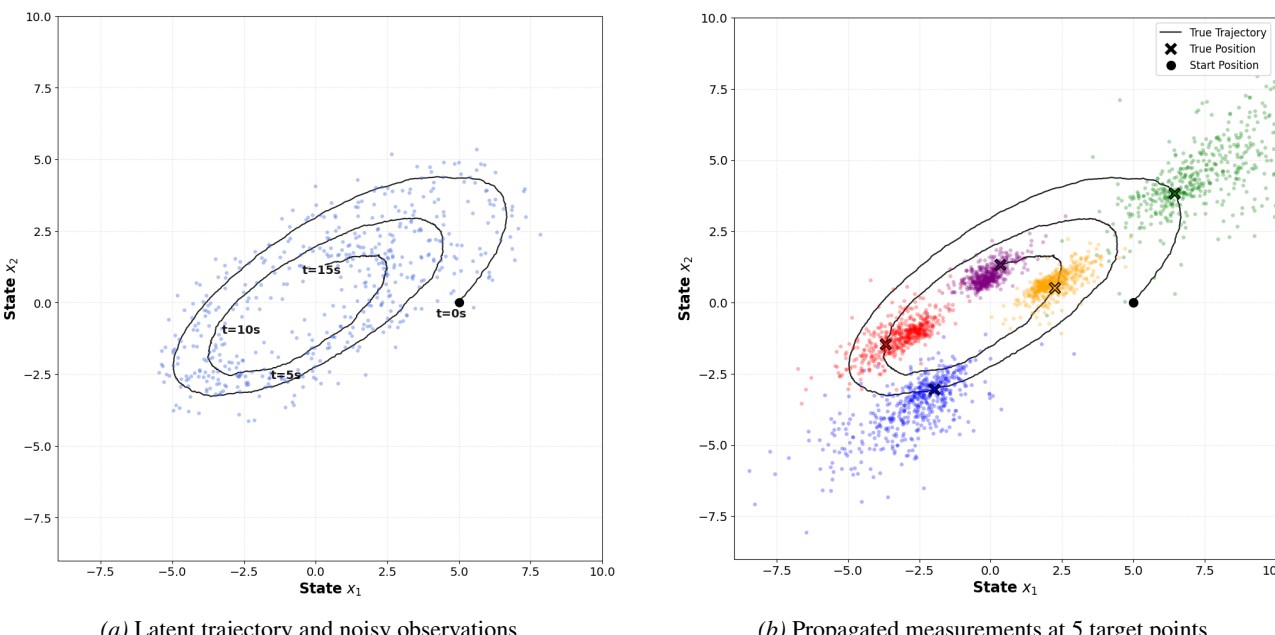

*(a)* Latent trajectory and noisy observations

*(b)* Propagated measurements at 5 target points

*Figure 1.* Illustration of an LTI SDE in 2D. **(a)** The true trajectory (black) is observed through additive Gaussian measurement noise $\boldsymbol{v}_t \sim \mathcal{N}(0, \boldsymbol{R})$; noisy measurements $\boldsymbol{z}_i$ are shown as blue points. **(b)** For five target times $t_i$, the plot shows the ensemble of transported observations $\hat{\boldsymbol{z}}_{ij}$ mapped through the transition $e^{\boldsymbol{A}\Delta t_{ij}}$ from all noisy measurements $\boldsymbol{z}_j$. Both forward and backward transported $\hat{\boldsymbol{z}}_{ij}$ are shown for visualization; estimation is causal and uses only $j \leq i$.

### A.1. Analytical Solution of the Differential Lyapunov Equation (DLE)

For parallel aggregation across all token pairs, we must efficiently solve the DLE for all $i, j \in [1, N]$. To obtain an analytically tractable solution, we assume the system matrices are simultaneously diagonalizable by an invertible $\boldsymbol{S} \in \mathbb{C}^{d \times d}$, where $\boldsymbol{A} = \boldsymbol{S}\boldsymbol{\Lambda}\boldsymbol{S}^{-1}$ and $\boldsymbol{Q} = \boldsymbol{S}\boldsymbol{\Lambda}_Q\boldsymbol{S}^\dagger$. This assumption corresponds to learning dynamics in a basis of decoupled modes.

The propagated state covariance, $\boldsymbol{V}(\Delta t_{ij})$, is the solution to the DLE (Eq. 2):

$$\boldsymbol{V}(\Delta t_{ij}) = \int_0^{\Delta t_{ij}} e^{\boldsymbol{A}s}\boldsymbol{Q}e^{\boldsymbol{A}^\top s}\,ds.$$

Transforming to the eigenbasis, the covariance becomes:

$$\boldsymbol{V}(\Delta t_{ij}) = \boldsymbol{S}\boldsymbol{\Lambda}_V(\Delta t_{ij})\boldsymbol{S}^\dagger.$$

where each diagonal entry of $\boldsymbol{\Lambda}_V(\Delta t_{ij})$ satisfies the scalar integral:

$$\lambda_{V,k}(\Delta t_{ij}) = \lambda_{Q,k} \int_0^{\Delta t_{ij}} e^{(\lambda_k + \lambda_k^*)s} ds = \lambda_{Q,k} \int_0^{\Delta t_{ij}} e^{2\mathrm{Re}(\lambda_k)s} ds$$

(where $\boldsymbol{\Lambda}_V = \mathrm{diag}(\lambda_V)$ and $\boldsymbol{\Lambda}_Q = \mathrm{diag}(\lambda_Q)$). Each mode accumulates noise according to its real decay rate $\mathrm{Re}(\lambda_k)$.

Evaluating this integral yields the analytical solution $\varphi(\lambda, \lambda_Q, \Delta t)$ (for the causal case $\Delta t \geq 0$):

$$\boldsymbol{\Lambda}_V(\Delta t_{ij}) = \mathrm{diag}\big(\varphi(\lambda_k, \lambda_{Q,k}, \Delta t_{ij})\big)_{k=1}^d,$$

$$\varphi(\lambda, \lambda_Q, \Delta t) = \begin{cases} \lambda_Q \dfrac{1 - e^{2\mathrm{Re}(\lambda)\,\Delta t}}{-2\,\mathrm{Re}(\lambda)}, & \mathrm{Re}(\lambda) \neq 0, \\ \lambda_Q\,\Delta t, & \mathrm{Re}(\lambda) = 0. \end{cases}$$

## A.2. Residual Covariance and Precision

The residual $\boldsymbol{r}_{ij} = \boldsymbol{z}_i - \hat{\boldsymbol{z}}_{ij}$ has covariance $\boldsymbol{\Sigma}_{ij} = \hat{\boldsymbol{V}}_{ij} + \boldsymbol{R}_\Gamma$ (Section 3.1). Assuming that $\boldsymbol{R} = \boldsymbol{S}\boldsymbol{\Lambda}_R\boldsymbol{S}^\dagger$ and $\boldsymbol{R}_\Gamma = \boldsymbol{S}\boldsymbol{\Lambda}_\Gamma\boldsymbol{S}^\dagger$, this becomes:

$$\boldsymbol{\Sigma}_{ij} = \boldsymbol{S}\boldsymbol{\Lambda}_{\Sigma,ij}\boldsymbol{S}^\dagger, \quad \boldsymbol{\Lambda}_{\Sigma,ij} = \boldsymbol{\Lambda}_V(\Delta t_{ij}) + e^{2\mathrm{Re}(\boldsymbol{\Lambda})\Delta t_{ij}}\boldsymbol{\Lambda}_R + \boldsymbol{\Lambda}_\Gamma.$$

This covariance is bounded for all $\Delta t_{ij} \geq 0$ if and only if $\mathrm{Re}(\lambda_k) < 0$ for all $k$, i.e., the dynamics are stable. The precision is then obtained by diagonal inversion:

$$\boldsymbol{P}_{ij} = \boldsymbol{S}^{-\dagger}\boldsymbol{\Lambda}_{P,ij}\boldsymbol{S}^{-1}, \qquad \boldsymbol{\Lambda}_{P,ij} = \boldsymbol{\Lambda}_{\Sigma,ij}^{-1}.$$

## A.3. Precision-Weighted Consensus State Estimation

To estimate the latent state at time $t_i$, we consider the transported observations $\{\hat{\boldsymbol{z}}_{ij}\}_{j \leq i}$, which are jointly Gaussian under the linear SDE with cross-covariances induced by shared process noise along the trajectory. The optimal estimator under this model requires inversion of a dense covariance matrix or sequential inference (e.g., Kalman filtering or smoothing), both of which retain temporal coupling and therefore do not yield the factorized attention form considered here.

Parallel aggregation requires factorizing the joint likelihood across observations. We adopt a mean-field (conditionally independent) approximation in which each transported observation is treated as an independent noisy measurement of the latent state, with its marginal covariance matched to the SDE-derived uncertainty.

Specifically, we treat the latent state $\boldsymbol{x}_i$ as a parameter $\boldsymbol{x} \in \mathbb{R}^d$ and model each transported observation as conditionally independent given $\boldsymbol{x}$:

$$p(\hat{\boldsymbol{z}}_{ij} \mid \boldsymbol{x}) = \mathcal{N}(\hat{\boldsymbol{z}}_{ij}; \boldsymbol{x}, \boldsymbol{\Sigma}_{ij}),$$

where $\boldsymbol{\Sigma}_{ij}$ is the SDE-derived marginal covariance of the residual, capturing both accumulated process noise and observation noise at the query position, and $\boldsymbol{P}_{ij} = \boldsymbol{\Sigma}_{ij}^{-1}$ is the corresponding precision.

Under this approximation, the likelihood factorizes:

$$p(\{\hat{\boldsymbol{z}}_{ij}\}_{j \leq i} \mid \boldsymbol{x}) \approx \prod_{j \leq i} p(\hat{\boldsymbol{z}}_{ij} \mid \boldsymbol{x}),$$

yielding the precision-weighted least-squares problem:

$$\bar{\boldsymbol{z}}_i = \arg\min_{\boldsymbol{x}} \sum_{j \leq i} (\boldsymbol{x} - \hat{\boldsymbol{z}}_{ij})^\top \boldsymbol{P}_{ij} (\boldsymbol{x} - \hat{\boldsymbol{z}}_{ij}).$$

Setting the gradient to zero gives the precision-weighted batch estimator:

$$\bar{z}_i = \left(\sum_{j\leq i} P_{ij}\right)^{-1} \sum_{j\leq i} P_{ij}\,\hat{z}_{ij}.$$

This approximation is exact when $Q = 0$. As process noise increases, uncertainty accumulates along the trajectory and induces correlations between transported observations, causing independent evidence to be over-counted. Nevertheless, the approximation preserves the correct marginal uncertainty for each transported observation, ensuring tokens are weighted according to their individual predicted reliability under the dynamics.

### A.4. Robust Precision Reweighting

The above estimator corresponds to a Gaussian likelihood over residuals. To account for model mismatch, we replace this with an M-estimator in which the precision prior $P_{ij}$ is reweighted by a scalar function of the Mahalanobis distance:

$$\tilde{P}_{ij} = w(d_{ij}^2)\,P_{ij}, \qquad d_{ij}^2 = r_{ij}^\top P_{ij} r_{ij}.$$

The reweighted estimator becomes:

$$\bar{z}_i = \left(\sum_{j\leq i} \tilde{P}_{ij}\right)^{-1} \sum_{j\leq i} \tilde{P}_{ij}\,\hat{z}_{ij}.$$

The choice of $w(\cdot)$ determines the robustness profile of the estimator. Two standard choices are an exponential, which corresponds to a Gaussian likelihood and recovers softmax-style attention, and a power-law, which corresponds to a Student-$t$ likelihood and provides heavier tails with increased robustness to outliers:

$$w_{ij} \propto \begin{cases} \exp\left(-\frac{d_{ij}^2}{\nu}\right) & \text{(exponential)} \\ \left(1 + \frac{d_{ij}^2}{\nu}\right)^{-\kappa} & \text{(power law)} \end{cases}$$

Since $d_{ij}^2$ depends on the unknown latent state, the weights $w_{ij}$ are implicitly functions of $\bar{z}_i$, yielding a fixed-point equation interpretable as one step of iteratively reweighted least squares (IRLS). This connection is developed in Appendix C.3, where stacked attention layers are interpreted as unrolling this iterative procedure across depth.

### A.5. Parallel Aggregation via Diagonalization

To obtain a scalable implementation, we transform the robust precision-weighted average to the diagonalized basis. We define the state and propagated measurements in this basis as:

$$z_{s,i} := S^{-1} z_i, \qquad \hat{z}_{s,ij} := e^{\Lambda \Delta t_{ij}}\,z_{s,j},$$

and the corresponding residual:

$$r_{s,ij} = z_{s,i} - \hat{z}_{s,ij}.$$

Using the simultaneous diagonalization, $P_{ij} = S^{-\dagger}\Lambda_{P,ij} S^{-1}$, and the Mahalanobis distance decomposes into independent scalar components:

$$d_{ij}^2 = r_{s,ij}^\dagger \Lambda_{P,ij} r_{s,ij} = \sum_{k=1}^{d} \lambda_{P,ij,k}\,\left|r_{s,ij,k}\right|^2.$$

where $\lambda_{P,ij,k}$ and $r_{s,ij,k}$ are the $k$th diagonal components of $\Lambda_{P,ij}$ and $r_{s,ij}$, respectively.

This allows the robust weights $w_{ij} = w(d_{ij}^2)$ to be computed efficiently for all token pairs.

Applying the aggregation in this basis yields:

$$\bar{z}_{s,i} = \left(\sum_{j\leq i} w_{ij}\,\Lambda_{P,ij}\right)^{-1} \sum_{j\leq i} w_{ij}\,\Lambda_{P,ij}\,\hat{z}_{s,ij}.$$

Since all matrices are diagonal, both the sum and inverse are element-wise operations.

Writing $\boldsymbol{\lambda}_{P,ij} := \operatorname{diag}(\boldsymbol{\Lambda}_{P,ij})$, we define attention weights as:

$$\tilde{\mathcal{A}}_{ij} := w_{ij}\,\boldsymbol{\lambda}_{P,ij}, \quad \mathcal{A}_{ij} = \tilde{\mathcal{A}}_{ij} \oslash \left( \sum_{j' \leq i} \tilde{\mathcal{A}}_{ij'} \right),$$

where $\oslash$ denotes element-wise division. The aggregation then takes the form:

$$\bar{\boldsymbol{z}}_{s,i} = \sum_{j \leq i} \mathcal{A}_{ij} \odot \hat{\boldsymbol{z}}_{s,ij},$$

where, instead of scalar attention weights, each $\mathcal{A}_{ij}$ is a vector that element-wise multiplies $\hat{\boldsymbol{z}}_{s,ij}$.

Finally, the output in the original coordinate system is recovered by $\bar{\boldsymbol{z}}_i = \boldsymbol{S}\,\bar{\boldsymbol{z}}_{s,i}$. All operations are $\mathcal{O}(d)$ per token pair, yielding an overall complexity of $\mathcal{O}(N^2 d)$ with no matrix inversions.

Equivalently, this normalization can be written in Softmax form by defining dimension-wise attention logits. For exponential reweighting $w_{ij} \propto \exp(-d_{ij}^2/\nu)$:

$$\mathcal{A}_{ij,k} = \frac{\exp(\mathcal{L}_{ij,k})}{\sum_{j' \leq i} \exp(\mathcal{L}_{ij',k})}, \quad \mathcal{L}_{ij,k} := \log \lambda_{P,ij,k} - \tfrac{1}{\nu}\,\lambda_{P,ij,k} \big| r_{s,ij,k} \big|^2.$$

For the power law influence function $w_{ij} \propto (1 + d_{ij}^2/\nu)^{-\kappa}$, the corresponding logit becomes:

$$\mathcal{L}_{ij,k} := \log \lambda_{P,ij,k} - \kappa \log \Big( 1 + \tfrac{1}{\nu}\lambda_{P,ij,k} \big| r_{s,ij,k} \big|^2 \Big),$$

In the isotropic case, where $\lambda_{P,ij,k} = \lambda_{P,ij}$ is a shared scalar across dimensions, both kernels reduce to scalar logits. In the exponential case,

$$\mathcal{L}_{ij} = \log(\lambda_{P,ij}) - \tfrac{1}{\nu}\lambda_{P,ij}\|\boldsymbol{r}_{s,ij}\|^2,$$

which is a Softmax normalization over scalar similarity scores. This shows that attention weights arise as normalized likelihood scores under the SDE-induced uncertainty model.

**Remark.** The normalization term $\left( \sum_{j' \leq i} \tilde{\mathcal{A}}_{ij'} \right)^{-1}$ corresponds to the approximate posterior covariance in the diagonalizing basis under the CI assumption, providing a measure of the model's confidence in the aggregated state estimate. This quantity is a byproduct of the formulation that goes unused in standard attention, and may be useful beyond weighting — for instance, to gate updates or signal when the context provides insufficient evidence. We leave exploration of such uses to future work.

### A.5.1. STUDENT-T LIKELIHOOD ATTENTION (ISOTROPIC CASE)

Under an isotropic covariance constraint $\boldsymbol{\Sigma}_{ij} = \Sigma^2(\Delta t_{ij})\boldsymbol{I}_d$, where $\Sigma^2(\Delta t_{ij}) \in \mathbb{R}^+$, the robust precision-reweighted attention score may be written as:

$$\mathcal{L}_{ij} = -\log\left(\Sigma^2(\Delta t_{ij})\right) - \kappa \log\Big( 1 + \tfrac{1}{\nu}d_{ij}^2 \Big),$$

where $d_{ij}^2 = \|\boldsymbol{r}_{ij}\|^2/\Sigma^2(\Delta t_{ij})$ is the isotropic Mahalanobis distance.

Choosing $\kappa = \frac{\nu+d}{d}$ makes this expression proportional (up to additive constants) to the dimension-normalized negative log-likelihood of an isotropic multivariate Student's $t$ distribution:

$$\mathcal{L}_{ij} = -\log\left(\Sigma^2(\Delta t_{ij})\right) + \tfrac{\nu+d}{d} \log\Big( 1 + \tfrac{1}{\nu}d_{ij}^2 \Big).$$

Setting $\nu = \nu_s d$, this becomes:

$$\mathcal{L}_{ij} = -\log\left(\Sigma^2(\Delta t_{ij})\right) + (\nu_s + 1) \log\Big( 1 + \tfrac{1}{\nu_s d}d_{ij}^2 \Big).$$

The dimension normalization is critical for stability: in high-dimensional spaces, squared Mahalanobis distances concentrate around their expected value of $d$, causing unnormalized likelihoods to produce overly sharp, near-deterministic weights. The exponent $\kappa = (\nu + d)/d$ is the unique value ensuring that the influence function treats a residual of typical size $d_{ij}^2 \sim d$ consistently regardless of dimension, preserving sensitivity to relative consistency rather than absolute norm. Under this choice, robust precision-weighted filtering is equivalent, up to additive constants that are absorbed by softmax normalization, to using dimension-normalized Student-$t$ log-likelihoods as attention logits.

# B. Robust Filter Attention Mechanism

We now instantiate the state estimation formulation as an attention mechanism, proceeding from the full anisotropic tensor formulation implied by the derivation to the memory-efficient isotropic variant used in practice.

## B.1. Anisotropic Tensor RFA

Under diagonalizable dynamics, the most general RFA formulation propagates and weights each feature dimension independently, yielding an $\mathcal{O}(N^2 d)$ attention tensor. This is not memory-scalable but serves as the reference from which the isotropic variant is derived.

**Learned Change-of-Basis Projections.** The transformation to the decoupled eigenbasis is learned through complex-valued projections. We define:

$$\boldsymbol{W}_q, \boldsymbol{W}_k, \boldsymbol{W}_v, \boldsymbol{W}_o \in \mathbb{C}^{d \times d},$$

where $d$ is the embedding dimension. The input projections $\{\boldsymbol{W}_q, \boldsymbol{W}_k, \boldsymbol{W}_v\}$ parameterize the learned diagonalizing basis $\boldsymbol{S}^{-1}$, mapping the input into the eigenbasis of $\boldsymbol{A}$ where the DLE is analytically solvable, while the output projection $\boldsymbol{W}_o$ parameterizes $\boldsymbol{S}$, mapping the filtered state estimate back into the original embedding space.

Given an input sequence $\boldsymbol{Z} \in \mathbb{R}^{d \times N}$, we obtain query, key, and value representations as usual:

$$\boldsymbol{Q} = \boldsymbol{W}_q \boldsymbol{Z}, \quad \boldsymbol{K} = \boldsymbol{W}_k \boldsymbol{Z}, \quad \boldsymbol{V} = \boldsymbol{W}_v \boldsymbol{Z}.$$

**Key and Value Propagation.** In this reference model, every feature $k$ possesses its own complex eigenvalue $\lambda_k$. We define the propagation tensors $\mathcal{E}$ and the resulting propagated keys and values:

$$\mathcal{E}[k,i,j] = e^{\boldsymbol{\lambda}_k(t_i - t_j)}, \qquad \hat{\mathcal{K}}[k,i,j] = \mathcal{E}[k,i,j] \cdot \boldsymbol{K}[k,j], \qquad \hat{\mathcal{V}}[k,i,j] = \mathcal{E}[k,i,j] \cdot \boldsymbol{V}[k,j].$$

**Residuals & Precision.** We compute the residual tensor $\mathcal{R}_{qk}$:

$$\mathcal{R}_{qk}[k,i,j] = \boldsymbol{Q}[k,i] - \hat{\mathcal{K}}[k,i,j],$$

This is weighted by the analytic precision tensor $\mathcal{P}$, which is defined element-wise for each channel $k$ using the DLE solution, where $\mu_k := -\mathrm{Re}(\lambda_k)$:

$$\mathcal{P}[k,i,j] = \left( \tilde{\sigma}_k^2 \left( 1 - e^{-2\mu_k \Delta t_{ij}} \right) + \eta_k^2 e^{-2\mu_k \Delta t_{ij}} + \gamma_k^2 \right)^{-1}.$$

**Aggregation.** Unlike standard attention, which applies a single scalar score per head, tensor RFA computes an attention tensor $\mathcal{A} \in \mathbb{R}^{d \times N \times N}$. The logit tensor is then:

$$\mathcal{L}[k,i,j] = \log \left( \mathcal{P}[k,i,j] \right) - \kappa \log \left( 1 + \frac{1}{\nu} \sum_{k'} \mathcal{P}[k',i,j] \cdot \left| \mathcal{R}_{qk}[k',i,j] \right|^2 \right).$$

The estimate in the eigenbasis is computed via a row-wise Softmax over the logits, followed by a weighted sum:

$$\mathcal{A}[k,i,j] = \mathrm{Softmax}_j(\mathcal{L}[k,i,j]), \quad \bar{\boldsymbol{V}}[k,i] = \sum_{j \leq i} \mathcal{A}[k,i,j] \cdot \hat{\mathcal{V}}[k,i,j].$$

The time complexity remains $\mathcal{O}(N^2 d)$, but storing the propagated keys and values, residuals, and attention tensors requires $\mathcal{O}(N^2 d)$ memory, limiting scalability. We therefore derive a memory-efficient implementation that avoids storing tensors.

## B.2. Factorization and Complexity Reduction

We introduce the following factorizations to simplify the computation:

### B.2.1. TOEPLITZ KERNEL FOR PRECISION

If the measurements occur at equal time intervals $\delta t$, the analytic precision kernel $\mathcal{P}[k, i, j]$ depends only on the channel $k$ and the time lag $\tau = |i - j|$. This induces a Toeplitz structure along the temporal dimensions for each channel.

Letting $\Delta t_{ij} = \tau \delta t$, we can thus pre-compute a 1D covariance kernel: $\mathcal{K}^\Sigma \in \mathbb{R}^{d \times N}$:

$$\mathcal{K}^\Sigma[k, \tau] = \tilde{\sigma}_k^2 \left(1 - e^{-2\mu_k \delta t \, \tau}\right) + \eta_k^2 \, e^{-2\mu_k \delta t \, \tau} + \gamma_k^2,$$

The full precision tensor is then simply the element-wise inverse of this kernel:

$$\mathcal{P}[k, i, j] = \mathcal{K}^P[k, |i - j|] := 1/\mathcal{K}^\Sigma[k, |i - j|].$$

### B.2.2. FACTORIZING THE PROPAGATED MEASUREMENTS

Because the dynamics are LTI, we can avoid decompose the propagation tensor $\mathcal{E}$ into separate forward and backward terms for each dimension $k$:

$$\mathcal{E}[k, i, j] = \mathbf{\Phi}^+[k, i] \cdot \mathbf{\Phi}^-[k, j], \quad \text{where:} \quad \mathbf{\Phi}^+[k, i] := e^{\boldsymbol{\lambda}_k t_i}, \quad \mathbf{\Phi}^-[k, i] := e^{-\boldsymbol{\lambda}_k t_i}.$$

We can then define stationary representations:

$$\hat{\mathbf{Q}}[k, j] := \mathbf{\Phi}^-[k, j] \cdot \mathbf{Q}[k, j], \quad \hat{\mathbf{K}}[k, j] := \mathbf{\Phi}^-[k, j] \cdot \mathbf{K}[k, j], \quad \hat{\mathbf{V}}[k, j] := \mathbf{\Phi}^-[k, j] \cdot \mathbf{V}[k, j].$$

The propagated keys and values are then refactored as products:

$$\hat{\mathcal{K}}[k, i, j] = \mathbf{\Phi}^+[k, i] \cdot \hat{\mathbf{K}}[k, j], \quad \hat{\mathcal{V}}[k, i, j] = \mathbf{\Phi}^+[k, i] \cdot \hat{\mathbf{V}}[k, j]$$

Since $\mathbf{\Phi}^+[k, i]$ does not depend on $j$, we can pull it outside the sum:

$$\bar{\mathbf{V}}[k, i] := \sum_{j \leq i} \mathcal{A}[k, i, j] \cdot \hat{\mathcal{V}}[k, i, j] = \mathbf{\Phi}^+[k, i] \cdot \sum_{j \leq i} \mathcal{A}[k, i, j] \cdot \hat{\mathbf{V}}[k, j]$$

### B.2.3. MEMORY EFFICIENCY AND NUMERICAL STABILITY

Recall that:

$$\mathcal{R}_{qk}[k, i, j] = \mathbf{Q}[k, i] - \hat{\mathcal{K}}[k, i, j]$$

Plugging in the factorizations for $\hat{\mathbf{Q}}[k, j]$ and $\hat{\mathcal{K}}[k, i, j]$, the residual becomes:

$$\mathcal{R}_{qk}[k, i, j] = \mathbf{Q}[k, i] - \mathbf{\Phi}^+[k, i] \cdot \hat{\mathbf{K}}[k, j] = \mathbf{\Phi}^+[k, i] \cdot \left(\hat{\mathbf{Q}}[k, i] - \hat{\mathbf{K}}[k, j]\right).$$

The matrix of Mahalanobis distances now becomes:

$$\mathbf{D}^2[i, j] = \sum_k \underbrace{\mathcal{K}^P[k, |i - j|]}_{\text{Precision kernel}} \cdot \underbrace{|\mathbf{\Phi}^+[k, i]|^2}_{\text{Forward decay}} \cdot \left[ \underbrace{\left|\hat{\mathbf{Q}}[k, i]\right|^2}_{\text{Stationary Query}} + \underbrace{\left|\hat{\mathbf{K}}[k, j]\right|^2}_{\text{Stationary Key}} - \underbrace{2 \operatorname{Re}\left(\hat{\mathbf{Q}}^*[k, i] \, \hat{\mathbf{K}}[k, j]\right)}_{\text{Stationary Cross-term}} \right]$$

(where $*$ denotes the complex conjugate). The remaining bottleneck is the $k$-dependence of the precision kernel $\mathcal{K}^P$ in the evaluation of the cross-term:

$$\sum_k \mathcal{K}^P[k, |i - j|] \cdot 2 \operatorname{Re}\left(\hat{\mathbf{Q}}^*[k, i] \, \hat{\mathbf{K}}[k, j]\right).$$

In standard attention, scores are computed with a single matrix multiplication ($\mathbf{Q}\mathbf{K}^\top$). Here, however, the precision kernel $\mathcal{K}^P[k, |i - j|]$ weights each feature differently as a function of time lag, so the summation over $k$ cannot be expressed as a single matmul. Achieving $\mathcal{O}(N^2 + Nd)$ memory therefore requires the precision kernel to be independent of the feature index $k$, allowing it to factor outside the summation.

A degenerate case occurs in the zero-noise limit, where the precision kernel is constant. This recovers a memory-efficient formulation with anisotropic (feature-wise) decay, similar to xPos.

However, for stable dynamics with $\mu_k > 0$, the backward transition factor $\boldsymbol{\Phi}^-[k,j] = e^{(\mu_k - i\omega_k)t_j}$ grows exponentially with sequence length. When decay rates vary across features, the stationary representations $\hat{\boldsymbol{Q}}, \hat{\boldsymbol{K}}, \hat{\boldsymbol{V}}$ grow exponentially with sequence length, making fully parallel computation numerically unstable because forward and backward factors cancel only after multiplication, allowing intermediate values to overflow.

Therefore, retaining a non-constant precision kernel while ensuring numerical stability under extrapolation requires restricting decay to be isotropic within each head. This allows decay to be factored at the head level rather than per feature, enabling stable, fully parallel attention with $\mathcal{O}(N^2 + Nd)$ memory. This motivates the Isotropic RFA variant introduced next.

### B.3. Isotropic RFA

#### B.3.1. ISOTROPIC DECAY AND NOISE ASSUMPTIONS

All assumptions in this section are applied *per attention head*. In particular, the real part of the eigenvalues within a head is taken to be a shared scalar $-\mu$:

$$\lambda_k = -\mu + i\omega_k, \qquad \mu \in \mathbb{R}^+, \omega_k \in \mathbb{R}$$

This corresponds to a system with an isotropic plus skew-symmetric state matrix:

$$\boldsymbol{A} = -\mu\boldsymbol{I} + \boldsymbol{\Omega}, \qquad \boldsymbol{\Omega} = -\boldsymbol{\Omega}^\top \in \mathbb{R}^{d\times d}.$$

We also assume that the noise is isotropic, i.e. that the noise covariances are scalar multiples of identity:

$$\boldsymbol{\Lambda}_Q = \sigma^2\boldsymbol{I}, \quad \boldsymbol{\Lambda}_R = \eta^2\boldsymbol{I}, \quad \boldsymbol{\Lambda}_\Gamma = \gamma^2\boldsymbol{I}$$

Under this constraint, the covariance kernel simplifies to a scalar function:

$$\Sigma^2(|i-j|) = \tilde{\sigma}^2\left(1 - e^{-2\mu\delta t|i-j|}\right) + \eta^2 e^{-2\mu\delta t|i-j|} + \gamma^2$$

Hence, the precision kernel becomes a scalar function of the time lag $\tau = |i-j|$, allowing it to be pulled outside the feature summation. Defining $\boldsymbol{\Sigma}_{\Delta t}[i,j] := \Sigma^2(|i-j|)$ and $\boldsymbol{P}_{\Delta t}[i,j] := 1/\boldsymbol{\Sigma}_{\Delta t}[i,j]$, the matrix of Mahalanobis distances become:

$$\boldsymbol{D}^2[i,j] = \boldsymbol{P}_{\Delta t}[i,j] \cdot \left(\sum_k \left|\mathcal{R}_{qk}[k,i,j]\right|^2\right) =: \boldsymbol{P}_{\Delta t}[i,j] \cdot \left\|\boldsymbol{R}_{qk}[i,j]\right\|^2,$$

(Note that $\left\|\boldsymbol{R}_{qk}\right\|$ denotes a matrix of vector norms, not a matrix norm.)

#### B.3.2. SIMPLIFYING THE SQUARED RESIDUAL NORM

The isotropic constraint allows the dynamics to be factored into a stable decay kernel and complex-valued forward/backward rotations:

$$\boldsymbol{E}[i,j] = e^{-\mu|t_i-t_j|}, \quad \tilde{\boldsymbol{\Phi}}^+[k,i] := e^{i\omega_k t_i}, \quad \tilde{\boldsymbol{\Phi}}^-[k,i] := e^{-i\omega_k t_i},$$

We can then define backward-rotated queries, keys, and values:

$$\tilde{\boldsymbol{Q}} := \tilde{\boldsymbol{\Phi}}^- \odot \boldsymbol{Q}, \quad \tilde{\boldsymbol{K}} := \tilde{\boldsymbol{\Phi}}^- \odot \boldsymbol{K}, \quad \tilde{\boldsymbol{V}} := \tilde{\boldsymbol{\Phi}}^- \odot \boldsymbol{V},$$

Note that:

$$\boldsymbol{\Phi}^+[k,i] = e^{-\mu t_i}\tilde{\boldsymbol{\Phi}}^+[k,i], \quad \hat{\boldsymbol{Q}}[k,i] = e^{\mu t_i}\tilde{\boldsymbol{Q}}[k,i], \quad \hat{\boldsymbol{K}}[k,j] = e^{\mu t_j}\tilde{\boldsymbol{K}}[k,j].$$

Plugging this into the expression for the Mahalanobis distance, and using the fact that complex rotation preserves magnitude:

$$\left\|\boldsymbol{R}_{qk}[i,j]\right\|^2 = \sum_k e^{-2\mu t_i}\left|\tilde{\boldsymbol{\Phi}}^+[k,i]\right|^2 \cdot \left[e^{2\mu t_i}\left|\tilde{\boldsymbol{Q}}[k,i]\right|^2 + e^{2\mu t_j}\left|\tilde{\boldsymbol{K}}[k,j]\right|^2 - 2e^{\mu t_i}e^{\mu t_j}\,\mathrm{Re}\big(\tilde{\boldsymbol{Q}}^*[k,i]\,\tilde{\boldsymbol{K}}[k,j]\big)\right].$$

$$= \sum_k \left[\left|\tilde{\boldsymbol{Q}}[k,i]\right|^2 + e^{-2\mu(t_i-t_j)}\left|\tilde{\boldsymbol{K}}[k,j]\right|^2 - 2e^{-\mu(t_i-t_j)}\,\mathrm{Re}\big(\tilde{\boldsymbol{Q}}[k,i]^*\,\tilde{\boldsymbol{K}}[k,j]\big).$$

Or, in vectorized form:

$$\left\|\boldsymbol{R}_{qk}[i,j]\right\|^2 = \underbrace{\|\boldsymbol{Q}_i\|^2}_{\text{Query Norm}} + \underbrace{\boldsymbol{E}[i,j]^2 \cdot \|\boldsymbol{K}_j\|^2}_{\text{Decayed Key Norm}} - \underbrace{2\,\boldsymbol{E}[i,j] \cdot \mathrm{Re}\big(\tilde{\boldsymbol{Q}}_i^\dagger \tilde{\boldsymbol{K}}_j\big)}_{\text{Propagated Cross-term}}$$

(since $\|\boldsymbol{Q}_i\|^2 = \|\tilde{\boldsymbol{Q}}_i\|^2$ and $\|\boldsymbol{K}_j\|^2 = \|\tilde{\boldsymbol{K}}_j\|^2$).

Hence, under the isotropic assumption, the cross-term $\mathrm{Re}(\tilde{\boldsymbol{Q}}^\dagger \tilde{\boldsymbol{K}})$ can be computed using one $\mathcal{O}(N^2 d)$ matrix multiplication, achieving the required memory efficiency.

### B.3.3. The Attention Matrix and Estimate

The logit matrix $\boldsymbol{L}$ is defined using a Student-t–inspired robust loss:

$$\boldsymbol{L} = \log\left(\boldsymbol{P}_{\Delta t}\right) - (\nu_s + 1)\log\left(1 + \frac{1}{\nu_s d}\boldsymbol{P}_{\Delta t} \odot \left\|\boldsymbol{R}_{qk}\right\|^2\right),$$

where $\nu_s \in \mathbb{R}^+$ and $d$ is the head dimension.

Defining a causal mask $\boldsymbol{M}_{\text{causal}}$ and adding an inverse temperature parameter $\beta_s$, we can then express the row-normalization using row-wise Softmax:

$$\boldsymbol{A}[i,j] = \text{Softmax}_j\left(\beta_s \boldsymbol{L}[i,j] + \boldsymbol{M}_{\text{causal}}\right).$$

The value aggregation is refactored for stability:

$$\bar{\boldsymbol{V}}[k,i] = \left(e^{-\mu t_i}\tilde{\boldsymbol{\Phi}}^+[k,i]\right) \cdot \sum_{j \leq i} \boldsymbol{A}[i,j] \cdot \left(e^{\mu t_j}\tilde{\boldsymbol{V}}[k,j]\right)$$

$$= \tilde{\boldsymbol{\Phi}}^+[k,i] \cdot \sum_{j \leq i}\left(\boldsymbol{A}[i,j] \cdot \boldsymbol{E}[i,j]\right) \cdot \tilde{\boldsymbol{V}}[k,j]$$

Hence, defining a decayed attention matrix $\hat{\boldsymbol{A}} := \boldsymbol{A} \odot \boldsymbol{E}$, the filtered estimate $\bar{\boldsymbol{V}}$ is computed by transforming the aggregation back into the original frame:

$$\bar{\boldsymbol{V}} = \tilde{\boldsymbol{\Phi}}^+ \odot \left(\tilde{\boldsymbol{V}}\hat{\boldsymbol{A}}^\top\right).$$

Or, in a form more typical for attention:

$$\bar{\boldsymbol{V}}^\top = (\tilde{\boldsymbol{\Phi}}^+)^\top \odot \left(\hat{\boldsymbol{A}}\tilde{\boldsymbol{V}}^\top\right)$$

This rotate–aggregate–counter-rotate structure aligns representations in a shared dynamical frame before aggregation (Fig. 2).

In practice, multi-head attention is implemented by applying this procedure independently across multiple heads, each with its own dynamical and uncertainty parameters. The resulting update steps are concatenated and projected in the standard Transformer manner.

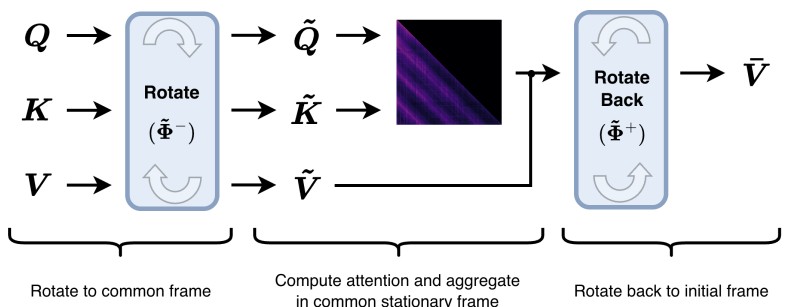

*Figure 2.* **Rotate, aggregate, counter-rotate structure of Isotropic RFA.** Queries, keys, and values are rotated into a common frame to compute attention and aggregate values. The resulting estimate is then rotated back to the initial frame, yielding a state that preserves relative phase while remaining equivariant to absolute position.

### B.3.4. Residual Connection and Output Projection

The filtered value $\bar{\boldsymbol{V}}$ should not be interpreted as a complete replacement for the current representation, but rather as a refined estimate under the robust filtering objective. Accordingly, the attention layer produces a correction relative to the current value representation:

$$\Delta\boldsymbol{V} := \bar{\boldsymbol{V}} - \boldsymbol{V}.$$

The residual connection then applies a partial step toward this estimate:

$$\boldsymbol{V}^+ = \boldsymbol{V} + \alpha \Delta \boldsymbol{V},$$

where $\alpha \in (0, 1]$ controls the step size. Mapping back to the original basis yields:

$$\boldsymbol{Z}^+ = \boldsymbol{W}_o \boldsymbol{V}^+ = \boldsymbol{W}_o \boldsymbol{V} + \alpha \boldsymbol{W}_o (\bar{\boldsymbol{V}} - \boldsymbol{V}).$$

Since the value projection maps residual representations into the latent dynamical space while the output projection maps the resulting corrections back into the residual stream, we expect $\boldsymbol{W}_o \boldsymbol{W}_v \approx \boldsymbol{I}$. Under this assumption, the update may be written directly in the residual stream:

$$\boldsymbol{Z}^+ \approx \boldsymbol{Z} + \alpha \boldsymbol{W}_o \Delta \boldsymbol{V}.$$

Hence, each attention layer contributes an incremental correction to the current representation, analogous to an iterative refinement step in state estimation.

### B.3.5. THE ZERO-DECAY LIMIT AND ALIBI

If the queries and keys are normalized, the matrix of squared residual norms becomes:

$$\left\| \boldsymbol{R}_{qk} \right\|^2 = \boldsymbol{1} + \boldsymbol{E}^2 - 2\,\boldsymbol{E} \odot \mathrm{Re}\left( \tilde{\boldsymbol{Q}}^\dagger \tilde{\boldsymbol{K}} \right).$$

In the zero-decay limit ($\mu \to 0$), the relative decay vanishes ($\boldsymbol{E} = \boldsymbol{1}$), and the residual simplifies to the chordal distance on the unit-norm hypersphere:

$$\left\| \boldsymbol{R}_{qk} \right\|^2 = 2\left( \boldsymbol{1} - \mathrm{Re}(\tilde{\boldsymbol{Q}}^\dagger \tilde{\boldsymbol{K}}) \right).$$

Substituting this into the NLL,

$$\boldsymbol{L} = \log(\boldsymbol{P}_{\Delta t}) - \kappa \log \left( 1 + \frac{2}{\nu} \boldsymbol{P}_{\Delta t} - \frac{2}{\nu} \boldsymbol{P}_{\Delta t} \odot \mathrm{Re}(\tilde{\boldsymbol{Q}}^\dagger \tilde{\boldsymbol{K}}) \right).$$

In the zero-decay limit, the covariance grows linearly with temporal lag:

$$\Sigma^2(\Delta t) = \sigma^2 \, \Delta t + (\eta^2 + \gamma^2), \quad \boldsymbol{P}_{\Delta t} = 1/\Sigma^2(\Delta t).$$

The resulting additive bias is dominated by the log-precision term :

$$\boldsymbol{B}_{\Delta t} \approx \log(\boldsymbol{P}_{\Delta t}) = -\log\left( \eta^2 + \gamma^2 \right) - \log\left( 1 + \frac{\sigma^2}{\eta^2 + \gamma^2} \Delta t \right).$$

For small $\Delta t$, this admits the linear approximation:

$$\boldsymbol{B}_{\Delta t} \approx -\log\left( \eta^2 + \gamma^2 \right) - \frac{\sigma^2}{\eta^2 + \gamma^2} \Delta t.$$

Thus, in the short-lag regime, the RFA prior induces an approximately linear distance-dependent bias, providing a local approximation to commonly used linear recency biases such as ALiBi.

### B.4. Diffusive and Integrative Filtering Regimes

Here we derive the explicit functional forms of the additive bias $\boldsymbol{B}_{\Delta t} := \log(\boldsymbol{P}_{\Delta t})$ and multiplicative gate $\boldsymbol{P}_{\Delta t}$ in each regime. The behavior of each regime is illustrated in Fig. 3, and the effect of $\mu$ on the speed of the phase transition is shown in Fig. 4.

**Diffusive Regime** ($\alpha < 0$). Letting $\alpha' = -\alpha > 0$, the bias follows a logarithmic decay:

$$\boldsymbol{B}_{\Delta t} = -\log(\beta) - \log\left( 1 - \frac{\alpha'}{\beta} e^{-2\mu \Delta t} \right),$$

starting at its maximum at $\Delta t = 0$ and decaying toward $-\log(\beta)$. The precision decays as:

$$\boldsymbol{P}_{\Delta t} = \left( \beta - \alpha' e^{-2\mu \Delta t} \right)^{-1},$$

with selectivity maximal near the diagonal and blurring out at longer lags.

**Integrative Regime ($\alpha > 0$).** The bias follows a mirrored Softplus:

$$\boldsymbol{B}_{\Delta t} = -\log(\beta) - \text{Softplus}(\ln(\alpha/\beta) - 2\mu\Delta t),$$

starting low and curving upward as key-side measurement noise dissipates. The precision follows a sigmoid:

$$\boldsymbol{P}_{\Delta t} = \frac{1}{\beta} \cdot \text{sigmoid}(2\mu\Delta t - \ln(\alpha/\beta)),$$

with selectivity initially low, opening as the transported observation becomes reliable.

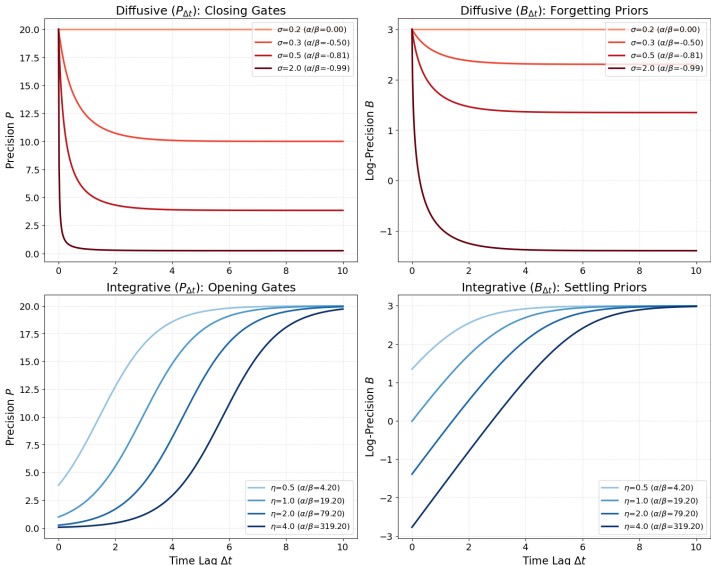

*Figure 3.* By varying the ratio of steady-state process uncertainty $\tilde{\sigma}^2$ to measurement noise $\eta^2$, RFA heads can specialize into distinct physical regimes: a **diffusive regime** that favors local recency (top row) and an **integrative regime** (bottom row) that filters transient noise to identify stable historical trends. The multiplicative gate $\boldsymbol{P}_{\Delta t}$ controls the selectivity (adaptive gain) of the attention, while the additive bias $\boldsymbol{B}_{\Delta t}$ defines the prior budget allocated to tokens at a given temporal lag.

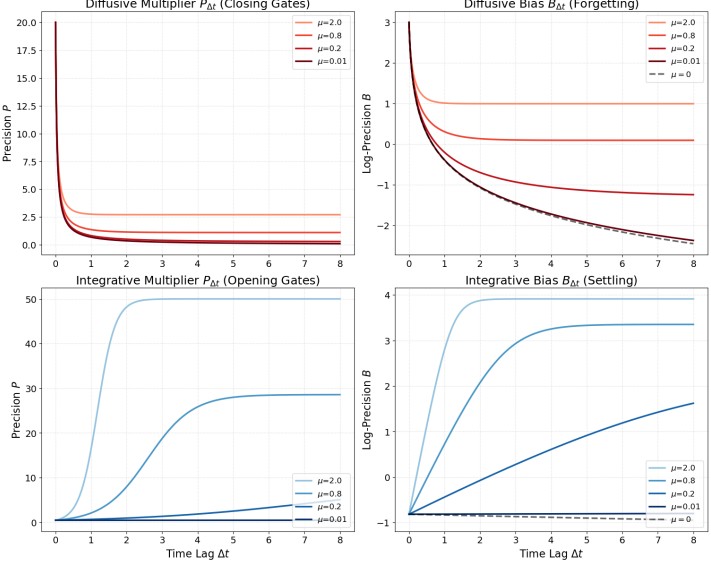

*Figure 4.* The decay rate $\mu$ dictates the speed of the phase transition. As $\mu \to 0$, the model recovers Brownian diffusion, where precision drops linearly with time. As $\mu$ increases, the model enforces stationarity, where the attention bias saturates to a learned global noise floor $\beta$, providing a principled mechanism for long-range context retention.

### B.5. Spectrally Coupled RFA

In standard RFA, all heads share the full frequency range with a uniform decay rate, so high- and low-frequency components are damped equally within each head (Fig. 5, left). SC-RFA partitions the spectrum across heads and couples decay rate to the maximum frequency per head (Fig. 5, right), inducing an ordered separation of temporal scales: high-frequency heads decay rapidly and act as short-range filters, while low-frequency heads decay slowly and preserve long-range structure. The resulting eigenvalue distribution in the complex plane is shown in Fig. 6.

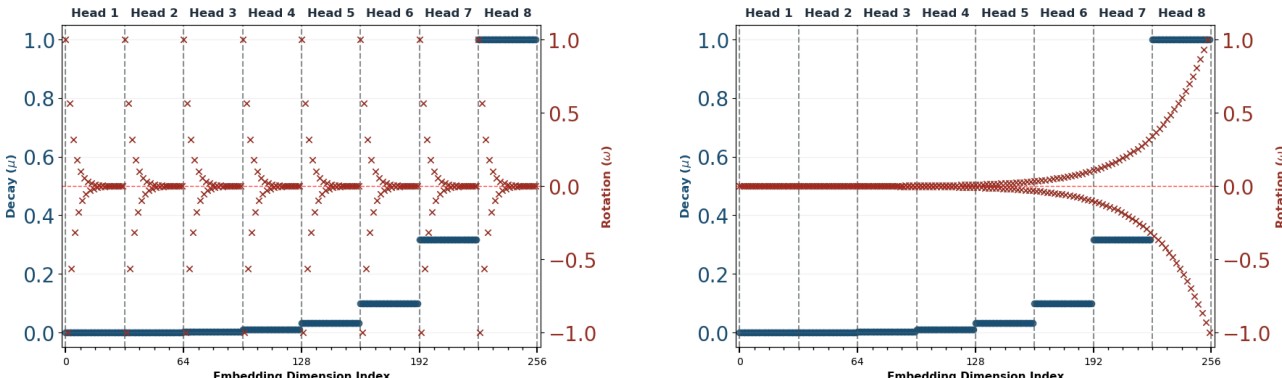

*Figure 5.* **Eigenvalue spectra of standard isotropic RFA (left) and SC-RFA (right), with** $b = 1.0$**.** Standard RFA uses the full frequency range with uniform decay per head. SC-RFA assigns each head a distinct spectral band with decay rate coupled to its maximum frequency.

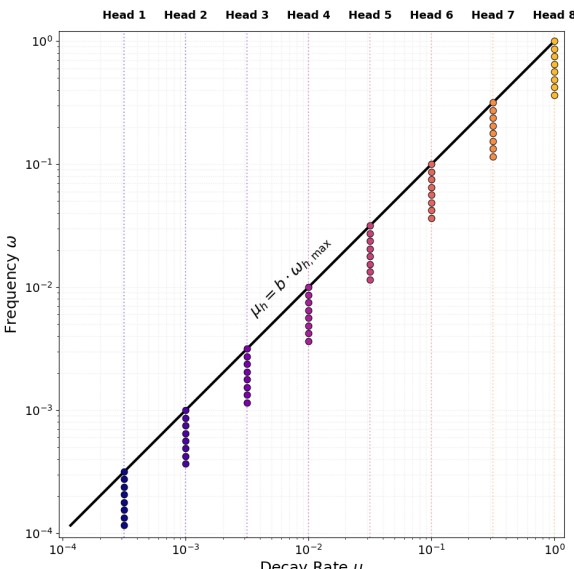

*Figure 6.* **SC-RFA eigenvalue distribution in log-log space** ($b = 1.0$)**.** The boundary $\mu_h = b \cdot \omega_{h,\max}$ appears as a straight line of slope $b$. Each head's eigenvalues form a vertical strip, with the highest-frequency eigenvalue on the boundary and lower-frequency eigenvalues falling below it.

## C. Extensions

This section presents some natural generalizations of the RFA framework.

### C.1. Inhomogeneous Dynamics

We show that constant drift does not alter the structure of RFA. Instead, it induces a deterministic shift in the propagated mean, which can be absorbed into learned bias terms in the query, key, and value projections.

Consider an inhomogeneous linear SDE with a constant drift $\boldsymbol{u}$:

$$d\boldsymbol{x}(t) = \big(\boldsymbol{A}\boldsymbol{x}(t) + \boldsymbol{u}\big)\, dt + \boldsymbol{G}\, d\boldsymbol{w}(t), \quad \boldsymbol{z}_k = \boldsymbol{x}(t_k) + \boldsymbol{v}(t_k).$$

The drift term can be eliminated via state augmentation, but this introduces a singular system that breaks simultaneous diagonalization. We therefore treat the inhomogeneous case directly.

We assume $\boldsymbol{A}$ is Hurwitz. Identifying the equilibrium point $\boldsymbol{\mu}$, the SDE can be rewritten as:

$$d\boldsymbol{x}(t) = \boldsymbol{A}\big(\boldsymbol{x}(t) - \boldsymbol{\mu}\big)\, dt + \boldsymbol{G}\, d\boldsymbol{w}(t), \qquad \boldsymbol{\mu} = -\boldsymbol{A}^{-1}\boldsymbol{u}.$$

The solution to the SDE, propagating the state forward from $\boldsymbol{x}(t_j)$ to $\boldsymbol{x}(t_i)$, is:

$$\boldsymbol{x}(t_i) = e^{\boldsymbol{A}\Delta t_{ij}}\boldsymbol{x}(t_j) + \left(\int_0^{\Delta t_{ij}} e^{\boldsymbol{A}(\Delta t_{ij}-\tau)}\, d\tau\right)\boldsymbol{u} + \int_0^{\Delta t_{ij}} e^{\boldsymbol{A}(\Delta t_{ij}-\tau)}\boldsymbol{G}\, d\boldsymbol{w}(\tau).$$

Letting $\boldsymbol{G}_u(\Delta t_{ij}) = \int_0^{\Delta t_{ij}} e^{-\boldsymbol{A}\tau}\, d\tau$, the deterministic part is:

$$\hat{\boldsymbol{x}}_{ij} = e^{\boldsymbol{A}\Delta t_{ij}}\Big(\boldsymbol{x}(t_j) + \boldsymbol{G}_u(\Delta t_{ij})\,\boldsymbol{u}\Big).$$

Letting $\boldsymbol{u}_s := \boldsymbol{S}^{-1}\boldsymbol{u}$, the drift term is:

$$\boldsymbol{G}_u(\Delta t_{ij})\boldsymbol{u} = \boldsymbol{S}\left(\int_0^{\Delta t_{ij}} e^{-\boldsymbol{\Lambda}\tau}\, d\tau\right)\boldsymbol{u}_s = \boldsymbol{S}\left(\frac{\boldsymbol{I} - e^{-\boldsymbol{\Lambda}\Delta t_{ij}}}{\boldsymbol{\Lambda}}\right)\boldsymbol{u}_s$$

Hence, the propagated measurement in the eigenbasis becomes:

$$\hat{\boldsymbol{z}}_{\boldsymbol{s},ij} = e^{\boldsymbol{\Lambda}\Delta t_{ij}}\left(\boldsymbol{z}_{s,j} + \left(\frac{\boldsymbol{I} - e^{-\boldsymbol{\Lambda}\Delta t_{ij}}}{\boldsymbol{\Lambda}}\right)\boldsymbol{u}_s\right) = e^{\boldsymbol{\Lambda}\Delta t_{ij}}\left(\boldsymbol{z}_{s,j} + \frac{\boldsymbol{u}_s}{\boldsymbol{\Lambda}}\right) - \frac{\boldsymbol{u}_s}{\boldsymbol{\Lambda}},$$

(where division is element-wise). Thus, drift induces a constant offset $-\boldsymbol{u}_s/\boldsymbol{\Lambda}$ in the diagonalized coordinates. Because the drift contributes only a deterministic shift, the covariance evolution remains identical to the homogeneous case.

Since this shift is constant across time, it can be absorbed into the learned linear projections by defining bias terms $\boldsymbol{b}_q, \boldsymbol{b}_k, \boldsymbol{b}_v \in \mathbb{C}^{d\times 1}$ in the input projections defining the queries, keys, and values:

$$\boldsymbol{Q}_u[k, i] := \boldsymbol{Q}[k, i] + \boldsymbol{b}_q[k], \quad \boldsymbol{K}_u[k, i] := \boldsymbol{K}[k, i] + \boldsymbol{b}_k[k], \quad \boldsymbol{V}_u[k, i] := \boldsymbol{V}[k, i] + \boldsymbol{b}_v[k],$$

$$\boldsymbol{b}_\ell[k] := \frac{\boldsymbol{u}_\ell[k]}{\boldsymbol{\lambda}_{\ell,k}}, \quad \ell \in \{q, k, v\},$$

where $\boldsymbol{u}_\ell$ and $\boldsymbol{\lambda}_{\ell,k}$ denote the $k$-th diagonal element of the drift and eigenvalue vector associated with the projection $\ell$. These bias terms correspond to the steady-state offset induced by constant drift in the diagonalized dynamics. This allows the residual tensor to maintain the same form as the homogeneous case, using the biased tensors:

$$\mathcal{R}_{qk}[k, i, j] = \boldsymbol{Q}_u[k, i] - \mathcal{E}_{qk}[k, i, j] \cdot \boldsymbol{K}_u[k, j].$$

The attention output is:

$$\bar{\boldsymbol{V}}[k, i] = \boldsymbol{\Phi}_v[k, i] \cdot \sum_{j\le i} \mathcal{A}[k, i, j] \cdot \hat{\boldsymbol{V}}_u[k, j] - \boldsymbol{b}_v[k] \cdot \sum_{j\le i} \mathcal{A}[k, i, j]$$

$$= \mathbf{\Phi}_v[k,i] \cdot \sum_{j \leq i} \mathcal{A}[k,i,j] \cdot \hat{\mathbf{V}}_u[k,j] - \mathbf{b}_v[k],$$

where we have used the fact that $\sum_{j \leq i} \mathcal{A}[k,i,j] = 1$ due to softmax normalization. The bias in this final expression can be absorbed into the bias of the output projection: $\mathbf{b}_o := \mathbf{W}_o \mathbf{b}_v$.

Hence, the inhomogeneous SDE with constant drift $\mathbf{u}$ is structurally equivalent to the homogeneous RFA mechanism, provided the deterministic effects are absorbed into constant bias vectors in the input and output projections ($\mathbf{b}_q, \mathbf{b}_k, \mathbf{b}_v, \mathbf{b}_o$).

## C.2. Generalized Analytic Priors via Time-Structured Noise

The derivation in Section A.1 assumed white process noise. However, each diagonal DLE is a linear ODE, and allowing the noise injection rate $q_k(t)$ to vary in time yields a richer class of analytic priors. For each mode $k$ with decay rate $\mu_k = -\mathrm{Re}(\lambda_k)$, the covariance satisfies:

$$\frac{d}{d\Delta t} \lambda_{V,k}(\Delta t) = -2\mu_k \lambda_{V,k}(\Delta t) + q_k(\Delta t), \qquad \lambda_{V,k}(0) = 0.$$

The unique solution is given by the convolution of the mode-specific noise source $q_k(s)$ with the system's exponential impulse response:

$$\lambda_{V,k}(\Delta t) = \int_0^{\Delta t} e^{-2\mu_k(\Delta t - s)} q_k(s)\, ds.$$

To ensure $\lambda_{V,k}(\Delta t)$ can be solved in closed-form, we restrict the noise source to the class of functions closed under exponential convolution: the complex exponentials. Letting $q_k(s) = \sum_j c_j e^{\gamma_j s}$ for $c_j, \gamma_j \in \mathbb{C}$, the integral yields a weighted sum of exponential differences:

$$\lambda_{V,k}(\Delta t) = \mathrm{Re}\left[ \sum_j c_j \left( \frac{e^{\gamma_j \Delta t} - e^{-2\mu_k \Delta t}}{2\mu_k + \gamma_j} \right) \right].$$

This characterizes the most general class of precision kernels that remain analytically tractable under a scalar DLE.

## C.3. Stacked Attention Layers as an Unrolled Iterative State Estimator

The robust M-estimator is defined implicitly: the weights $w_{ij}$ depend on residuals computed against the unknown latent state $\mathbf{x}_i$, which must itself be approximated from the current iterate. Each attention layer can therefore be interpreted as one step of an Iteratively Reweighted Least Squares (IRLS)-like procedure: given the previous layer's state estimate, the current layer recomputes residuals, updates weights, and produces a refined precision-weighted average.

Working in the eigenbasis $\mathbf{z}_{s,i} := \mathbf{S}^{-1} \mathbf{z}_i$, the procedure is initialized with each position's own embedding as the zeroth estimate: $\hat{\mathbf{z}}_{s,ii}^{(1)} = \mathbf{z}_{s,i}$. At each iteration, transported predictions are recomputed from the current state estimates:

$$\hat{\mathbf{z}}_{s,ij}^{(k)} = e^{\mathbf{\Lambda} \Delta t_{ij}} \hat{\mathbf{z}}_{s,jj}^{(k)}.$$

Weights are recomputed from the Mahalanobis residuals:

$$w_{ij}^{(k)} := \left( 1 + (\mathbf{\lambda}_{P,ij})^\top |\mathbf{r}_{s,ij}^{(k)}|^2 / \nu \right)^{-\kappa}, \qquad \mathbf{r}_{s,ij}^{(k)} := \hat{\mathbf{z}}_{s,ii}^{(k)} - \hat{\mathbf{z}}_{s,ij}^{(k)}.$$

A single refinement step $k$ computes the precision-weighted estimate

$$\bar{\mathbf{z}}_{s,i}^{(k)} = \left( \sum_{j \leq i} w_{ij}^{(k)} \mathbf{\lambda}_{P,ij} \right)^{-1} \odot \sum_{j \leq i} w_{ij}^{(k)} \mathbf{\lambda}_{P,ij} \odot \hat{\mathbf{z}}_{s,ij}^{(k)}.$$

where $\mathbf{\lambda}_{P,ij} := \mathbf{\lambda}_P(\Delta t_{ij})$ is the diagonal precision vector at lag $\Delta t_{ij}$ (Appendix A).

The estimate may then be updated via an innovation step:

$$\hat{\mathbf{z}}_{s,ii}^{(k+1)} = \hat{\mathbf{z}}_{s,ii}^{(k)} + \alpha_i \left( \bar{\mathbf{z}}_{s,i}^{(k)} - \hat{\mathbf{z}}_{s,ii}^{(k)} \right),$$

where $\alpha_i \in (0,1]$ controls the correction step size. This mirrors the residual updates used in standard Transformers, which may similarly be interpreted as iterative correction steps. Stacking $L$ attention layers with shared parameters can be interpreted as unrolling $L$ steps of this iterative estimation procedure.

# D. Implementation

## D.1. Complex-valued Computations

RFA is formulated in a complex latent space. In practice, this is implemented by lifting real-valued representations into a $2d$-dimensional space using a linear projection (corresponding to $\mathbb{C}^d \cong \mathbb{R}^{2d}$), performing complex rotations and attention in this space, and then projecting the result back to $\mathbb{R}^d$.

A complex-valued linear transformation can be represented in the real domain by operating on paired real and imaginary channels. For an input $\boldsymbol{x} = [\boldsymbol{x}_r, \boldsymbol{x}_i]^\top$ with $\boldsymbol{x}_r, \boldsymbol{x}_i \in \mathbb{R}^d$, this corresponds to:

$$\mathcal{L}(\boldsymbol{x}) = \begin{bmatrix} \boldsymbol{W}_r & -\boldsymbol{W}_i \\ \boldsymbol{W}_i & \boldsymbol{W}_r \end{bmatrix} \boldsymbol{x} + \begin{bmatrix} \boldsymbol{b}_r \\ \boldsymbol{b}_i \end{bmatrix},$$

Here $\boldsymbol{W}_r, \boldsymbol{W}_i \in \mathbb{R}^{d \times d}$ are the real and imaginary components of the weight matrix and $\boldsymbol{b}_r, \boldsymbol{b}_i \in \mathbb{R}^d$ the bias. This is equivalent to multiplication by a complex matrix $\boldsymbol{W} = \boldsymbol{W}_r + i\boldsymbol{W}_i$ with bias $\boldsymbol{b} = \boldsymbol{b}_r + i\boldsymbol{b}_i$.

Assuming the inputs and outputs are purely real, only the real-input columns of the input projections and the real-output columns of the output projections are required:

$$\mathcal{L}^{d \times 2d}(\boldsymbol{x}_r) := \begin{bmatrix} \boldsymbol{W}_r \\ \boldsymbol{W}_i \end{bmatrix} \boldsymbol{x}_r + \begin{bmatrix} \boldsymbol{b}_r \\ \boldsymbol{b}_i \end{bmatrix}.$$

$$\mathcal{L}^{2d \times d}(\boldsymbol{x}) := \begin{bmatrix} \boldsymbol{W}_r & -\boldsymbol{W}_i \end{bmatrix} \boldsymbol{x} + \boldsymbol{b}_r.$$

Hence, both projections may be implemented using standard real-valued linear layers in $\mathbb{R}^{2d}$.

We define queries, keys, and values using:

$$\boldsymbol{Q} = \mathcal{L}_q^{d \times 2d}(\boldsymbol{Z}), \quad \boldsymbol{K} = \mathcal{L}_k^{d \times 2d}(\boldsymbol{Z}), \quad \boldsymbol{V} = \mathcal{L}_v^{d \times 2d}(\boldsymbol{Z})$$

We define cosine and sine matrices:

$$\boldsymbol{C}[k, i] = \cos(\omega_k t_i), \quad \boldsymbol{S}[k, i] = \sin(\omega_k t_i)$$

Complex rotations are applied as:

$$\tilde{\boldsymbol{Q}}^\top = \begin{bmatrix} \tilde{\boldsymbol{Q}}_r^\top \\ \tilde{\boldsymbol{Q}}_i^\top \end{bmatrix} = (\tilde{\boldsymbol{\Phi}}^-)^\top \odot \boldsymbol{Q}^\top = \begin{bmatrix} \boldsymbol{C} \odot \boldsymbol{Q}_r^\top + \boldsymbol{S} \odot \boldsymbol{Q}_i^\top \\ \boldsymbol{C} \odot \boldsymbol{Q}_i^\top - \boldsymbol{S} \odot \boldsymbol{Q}_r^\top \end{bmatrix},$$

and likewise for $\tilde{\boldsymbol{K}}$ and $\tilde{\boldsymbol{V}}$. This is algebraically identical to RoPE.

To ensure the underlying system matrix $\boldsymbol{A}$ is real-valued, we enforce that its eigenvalues appear in complex conjugate pairs:

$$\boldsymbol{\omega} = \{\omega_1, -\omega_1, \ldots, \omega_{d/2}, -\omega_{d/2}\},$$

The Mahalanobis distance requires the real part of the complex inner product,

$$\mathrm{Re}\left(\tilde{\boldsymbol{Q}}^\dagger \tilde{\boldsymbol{K}}\right) = \tilde{\boldsymbol{Q}}_r^\top \tilde{\boldsymbol{K}}_r + \tilde{\boldsymbol{Q}}_i^\top \tilde{\boldsymbol{K}}_i = \begin{bmatrix} \tilde{\boldsymbol{Q}}_r^\top & \tilde{\boldsymbol{Q}}_i^\top \end{bmatrix} \begin{bmatrix} \tilde{\boldsymbol{K}}_r \\ \tilde{\boldsymbol{K}}_i \end{bmatrix}.$$

This is implemented as a single real matrix multiplication in $\mathbb{R}^{2d}$.

Value aggregation, $\bar{\boldsymbol{V}}$, is computed in the $\mathbb{R}^{2d}$ domain. The real-valued attention matrix $\hat{\boldsymbol{A}}$ is applied identically to both the real and imaginary components of the complex-rotated values:

$$\boldsymbol{M}^\top = \hat{\boldsymbol{A}} \begin{bmatrix} \tilde{\boldsymbol{V}}_r^\top \\ \tilde{\boldsymbol{V}}_i^\top \end{bmatrix}.$$

The inverse rotation yields:

$$\bar{\boldsymbol{Z}}_v^\top = (\tilde{\boldsymbol{\Phi}}^+)^\top \odot (\hat{\boldsymbol{A}}\tilde{\boldsymbol{V}}^\top) = \begin{bmatrix} \boldsymbol{C} \odot \boldsymbol{M}_r^\top - \boldsymbol{S} \odot \boldsymbol{M}_i^\top \\ \boldsymbol{C} \odot \boldsymbol{M}_i^\top + \boldsymbol{S} \odot \boldsymbol{M}_r^\top \end{bmatrix}$$

The final output is projected back to the real domain using the $\mathcal{L}^{2d \times d}$ layer:

$$\bar{\boldsymbol{Z}} = \mathcal{L}_o^{2d \times d}(\bar{\boldsymbol{V}}) = \begin{bmatrix} \boldsymbol{W}_r & -\boldsymbol{W}_i \end{bmatrix} \bar{\boldsymbol{V}} + \boldsymbol{b}_r \in \mathbb{R}^{d \times N}.$$

All components of RFA are therefore implemented using standard real-valued operations.

## D.2. Initialization

**Isotropic Complex Projections.** Complex weights $\boldsymbol{W} = \boldsymbol{W}_r + i\boldsymbol{W}_i$ are initialized isotropically:

$$\boldsymbol{W}_{ij} = M_{ij} \begin{bmatrix} \cos(\phi_{ij}) \\ \sin(\phi_{ij}) \end{bmatrix}, \quad M_{ij} \sim \text{Rayleigh}\left(\sqrt{\frac{1}{d_{\text{in}} + d_{\text{out}}}}\right), \quad \phi_{ij} \sim \mathcal{U}(0, 2\pi).$$

Output projections ($\boldsymbol{W}_o$) are scaled by $1/\sqrt{2}$ to preserve variance when converting back to real space.

**Noise and Robustness.** We initialize a constant steady-state uncertainty $\tilde{\sigma}$ across heads by scaling the process noise with the decay rate, $\sigma = 0.1\,\mu$. This ensures that the variance floor remains comparable across heads despite differences in temporal persistence. We initialize measurement noise ($\eta^2, \gamma^2$) such that such that the model begins in the integrative regime ($\eta^2 > \tilde{\sigma}^2$), to preserve long-range gradient flow early in training. We enforce positive query-side measurement noise $\gamma^2 > 0$ to ensure finite precision.

The Student-$t$ degrees of freedom $\nu$ are initialized as a positive multiple of the head dimension: $\nu = \nu_s d$. We initialized $\nu_s = 4$, placing the model in a quasi-Gaussian regime during the initial phase of training. This provides a broad prior that prevents the premature rejection of tokens while the Query-Key representations are still unoptimized.

**Remark.** In our implementation, $\sigma^2$ was learned directly. An equivalent and often more numerically stable parameterization is obtained by learning the steady-state variance $\tilde{\sigma}^2 := \sigma^2/(2\mu)$ directly. This decouples the variance floor from the decay rate $\mu$, improving conditioning when $\mu$ varies across heads. We drop the update step size $\alpha$, since it may be absorbed into the output projection $\boldsymbol{W}_o$.

## D.3. Algorithm

Algorithm 1 details the implementation of Isotropic RFA.

(**Note:** We use $\oplus$ to denote broadcast addition.)

---

**Algorithm 1** Robust Filter Attention (Isotropic; Single Head)

---

**Input:** Input sequence $\boldsymbol{Z} \in \mathbb{R}^{d \times N}$

**Definitions:**
**Linear layers:** $\mathcal{L}_q^{d \times 2d}, \mathcal{L}_k^{d \times 2d}, \mathcal{L}_v^{d \times 2d}, \mathcal{L}_o^{2d \times d}$.

**Scalar parameters:** Noise variance parameters: $\sigma', \eta', \gamma'$; robustness parameter $\nu_s$; Softmax inverse temperature $\beta_s$.

**Constants:** Causal mask $\boldsymbol{M}_{\text{causal}} \in \{0, -\infty\}^{N \times N}$; angular frequencies $\boldsymbol{\omega}$; decay rate $\mu \in \mathbb{R}^+$.

**Enforce Conjugate Symmetry:** $\boldsymbol{\omega} \in \{\omega_1', -\omega_1', \ldots, \omega_{d/2}', -\omega_{d/2}'\}$.

**Ensure positive noise/decay parameters:**
$\{\tilde{\sigma}^2, \eta^2, \gamma^2\} \leftarrow \text{Softplus}(\{\sigma', \eta', \gamma'\})$

**Input projections:**
$(\text{Re}(\boldsymbol{Q}), \text{Im}(\boldsymbol{Q})) \leftarrow \mathcal{L}_q(\boldsymbol{Z})$
$(\text{Re}(\boldsymbol{K}), \text{Im}(\boldsymbol{K})) \leftarrow \mathcal{L}_k(\boldsymbol{Z})$
$(\text{Re}(\boldsymbol{V}), \text{Im}(\boldsymbol{V})) \leftarrow \mathcal{L}_v(\boldsymbol{Z})$

**Decay and rotation kernels:** $\boldsymbol{E}[i,j] = e^{-\mu|t_i - t_j|}, \quad \tilde{\boldsymbol{\Phi}}^+[k,i] = e^{i\boldsymbol{\omega}_k t_i}, \quad \tilde{\boldsymbol{\Phi}}^-[k,i] = e^{-i\boldsymbol{\omega}_k t_i}$

**Covariance kernel:** $\boldsymbol{\Sigma}_{\Delta t}[i,j] = \tilde{\sigma}^2\left(1 - \boldsymbol{E}[i,j]^2\right) + \eta^2 \boldsymbol{E}[i,j]^2 + \gamma^2$

**Query/Key/Value Rotations:** $\tilde{\boldsymbol{Q}}[k,i] = \tilde{\boldsymbol{\Phi}}^- \odot \boldsymbol{Q}[k,i], \quad \tilde{\boldsymbol{K}}[k,j] = \tilde{\boldsymbol{\Phi}}^- \odot \boldsymbol{K}[k,j] \quad \tilde{\boldsymbol{V}}[k,i] = \tilde{\boldsymbol{\Phi}}^- \odot \boldsymbol{V}[k,i]$

**Squared residuals:** $\|\boldsymbol{R}_{qk}[i,j]\|^2 = \|\boldsymbol{Q}_i\|^2 + \boldsymbol{E}[i,j]^2 \cdot \|\boldsymbol{K}_j\|^2 - 2\boldsymbol{E}[i,j] \cdot \text{Re}(\tilde{\boldsymbol{Q}}_i^\dagger \tilde{\boldsymbol{K}}_j)$

**Logits:** $\boldsymbol{L} = -\log(\boldsymbol{\Sigma}_{\Delta t}) - (\nu_s + 1)\log\left(1 + \frac{1}{\nu_s d}\|\boldsymbol{R}_{qk}\|^2 \oslash \boldsymbol{\Sigma}_{\Delta t}\right)$.

**Attention matrix:** $\boldsymbol{A}[i,j] = \text{Softmax}_j\left(\beta_s \boldsymbol{L}[i,j] + \boldsymbol{M}_{\text{causal}}\right), \quad \hat{\boldsymbol{A}} = \boldsymbol{A} \odot \boldsymbol{E}$

**Value estimate:** $\bar{\boldsymbol{V}} = \tilde{\boldsymbol{\Phi}}^+ \odot (\tilde{\boldsymbol{V}} \hat{\boldsymbol{A}}^\top)$

**Value step:** $\Delta \boldsymbol{V} = \bar{\boldsymbol{V}} - \boldsymbol{V}$

**Output projection:** $\Delta \boldsymbol{Z} \leftarrow \mathcal{L}_o(\Delta \boldsymbol{V})$

**Residual connection:** $\boldsymbol{Z}^+ = \boldsymbol{Z} + \Delta \boldsymbol{Z}$

**Return:** $\boldsymbol{Z}^+$

---

Our current implementation is written in high-level PyTorch and incurs an approximately $2\times$ training overhead relative to PyTorch's optimized scaled dot-product attention backend; we expect this gap to be reduced with kernel fusion and optimized implementations.

# E. Experimental Details and Ablations

### E.1. Experimental Setup

**Architecture and Model Configuration.** All experiments were conducted using a 6-layer decoder-only Transformer architecture. We set the model dimension to $d_{\text{model}} = 256$ with $h = 8$ attention heads. The attention mechanism maps the model dimension to a total latent dimension of $512$ via the $d \times 2d$ query, key, and value projections (split into $d_h = 64$ per head), while the $2d \times d$ output projection maps back down to $256$.

We employ a Pre-Norm configuration using Layer Normalization. The Feed-Forward Network utilizes an expansion factor of 4. To optimize the parameter budget, we implement weight tying between the token embedding layer and the final linear output head. We use the GPT-2 byte-pair encoding (BPE) tokenizer with a vocabulary size of 50,257 for all language modeling experiments.

To ensure a fair comparison, RFA models and the baselines (RoPE and ALiBi) were designed with near-identical parameter counts. RFA introduces only a small set of scalar coefficients per head to parameterize noise variances ($\tilde{\sigma}^2, \eta^2, \gamma^2$) and robustness ($\nu, \beta_s$). Hence, the RFA models match the baseline parameter count (19.36M), with only a 0.02% increase due to additional scalar coefficients.

**Training and Optimization Protocol.** Models were trained for 15 epochs using the Adam optimizer. We utilized a OneCycleLR scheduler with cosine annealing and a 5% warmup period, and trained until convergence. For RFA models, we adopted a decoupled optimization strategy to ensure the stability of the SDE coefficients:

Feature weights use a peak LR of $1 \times 10^{-3}$ with $\beta_1 = 0.9$, while SDE coefficients use $5 \times 10^{-4}$ with $\beta_1 = 0.0$ and $\epsilon = 10^{-7}$. We apply global gradient clipping at 1.0, with a stricter $1 \times 10^{-4}$ threshold for RFA-specific parameters.

All models were trained on the WikiText-103 and BabyLM-2025 datasets using a standard causal language modeling objective.

### E.2. Model Variants and Ablation Design

This section presents a series of ablations designed to isolate the contributions of RFA's core components. The ablations consist of:

1. Baseline models;

2. RFA variants evaluated in Section 4 (M1–M3); and

3. Structural diagnostic ablations (M2.1–M2.6), which progressively remove components of the filtering formulation to test necessity and failure modes. These models are not intended as competitive models, but rather as mechanistic probes of stability and extrapolation behavior.

**Baselines:**

- **B1: Standard Transformer + RoPE.** Dot-product attention with rotary positional embeddings (Su et al., 2024). Applies $d \rightarrow 2d \rightarrow d$ projections to match RFA parameterization.

- **B2: Standard Transformer + ALiBi.** Dot-product attention with linear distance bias (Press et al., 2022). Tests whether static geometric penalties are sufficient for stability. Applies $d \rightarrow 2d \rightarrow d$ projections to match RFA parameterization.

- **B3: Decayed RoPE.** Identical to B1 but with an additional exponential decay applied to attention scores per-head, as in RFA, testing whether decay alone suffices in the absence of uncertainty modeling.

- **B4: Spectrally Coupled RoPE (SC-RoPE).** Identical to B3 but with frequency-partitioned RoPE with head-wise decay schedules, testing whether decay with spectral coupling can recover SC-RFA's stability.

**Primary RFA Models:**

- **M1: Isotropic RFA.** Isotropic RFA as described in Algorithm D.3, replacing the attention module in a standard Transformer. The first two heads are reserved with $\mu_h = 0$, and $\tilde{\sigma}_h$ is initialized to $0.1\mu_h$.

- **M2: Spectrally Coupled RFA (SC-RFA), optimized for near-field performance.** Identical to M1 except with explicit coupling between rotation frequencies and decay rates, $\mu_h = b \cdot \omega_{h,\max}$, using light damping ($b = 0.05$).

- **M3: Spectrally Coupled RFA (SC-RFA), optimized for extrapolation.** Identical to M2 but with stronger damping ($b = 5.0$), and $\tilde{\sigma}_h$ initialized to $0.5\mu_h$.

**Structural Diagnostic Ablations:** These ablations progressively remove components of the filtering formulation, starting from the full SC-RFA model (M2) and simplifying toward standard attention. Their purpose is to isolate which mechanisms are required for stable extrapolation.

- **M2.1: Exponential Kernel.** M2 with Student's $t$ influence function replaced by an exponential weighting, i.e., $w_{ij} = \exp(-d_{ij}^2/\nu)$. This isolates the effect of heavy-tailed robust reweighting under the same dynamical precision prior.

- **M2.2: Flat Precision Prior.** M2 with noise parameters removed so that $\boldsymbol{P}_{\Delta t}$ is constant across time lag. Tests whether dynamics alone suffice without uncertainty accumulation.

- **M2.3: No Multiplicative Gate.** M2 with the multiplicative gating term $\boldsymbol{P}_{\Delta t}$ set to a constant, to test the impact of the additive bias $\boldsymbol{B}_{\Delta t}$ in isolation.

- **M2.4: No Value Frame Alignment.** M2 without value rotation and counter-rotation, testing the necessity of aggregating in a shared temporal frame.

- **M2.5: No Rotational Dynamics.** M2 without rotations applied to queries, keys, or values, isolating the effect of decay-only dynamics.

- **M2.6: Unitary Dot-Product Limit.** No decay, no process or measurement noise, and no query and key normalization terms, so that attention weights reduce to normalized complex dot products between rotated embeddings. This yields a purely rotational positional encoding analogous to RoPER (Harik & Jayasiri, 2022).

# F. Additional Experimental Results

## F.1. Training Dynamics and Extrapolation Behavior

To assess learning efficiency, we track validation perplexity throughout training (Fig. 7a). All RFA variants (M1-M3) achieve consistently lower validation perplexity than the baselines (B1 and B2) over the course of training. This suggests that the SDE-based prior provides a more informative inductive bias than purely geometric positional encodings.

Figure 7b visualizes validation PPL and length extrapolation trends corresponding to the tabulated results in Section 4. RFA variants converge faster and degrade more gradually with context length than RoPE, while ALiBi remains stable due to its enforced locality, at the cost of worse performance within the training window.

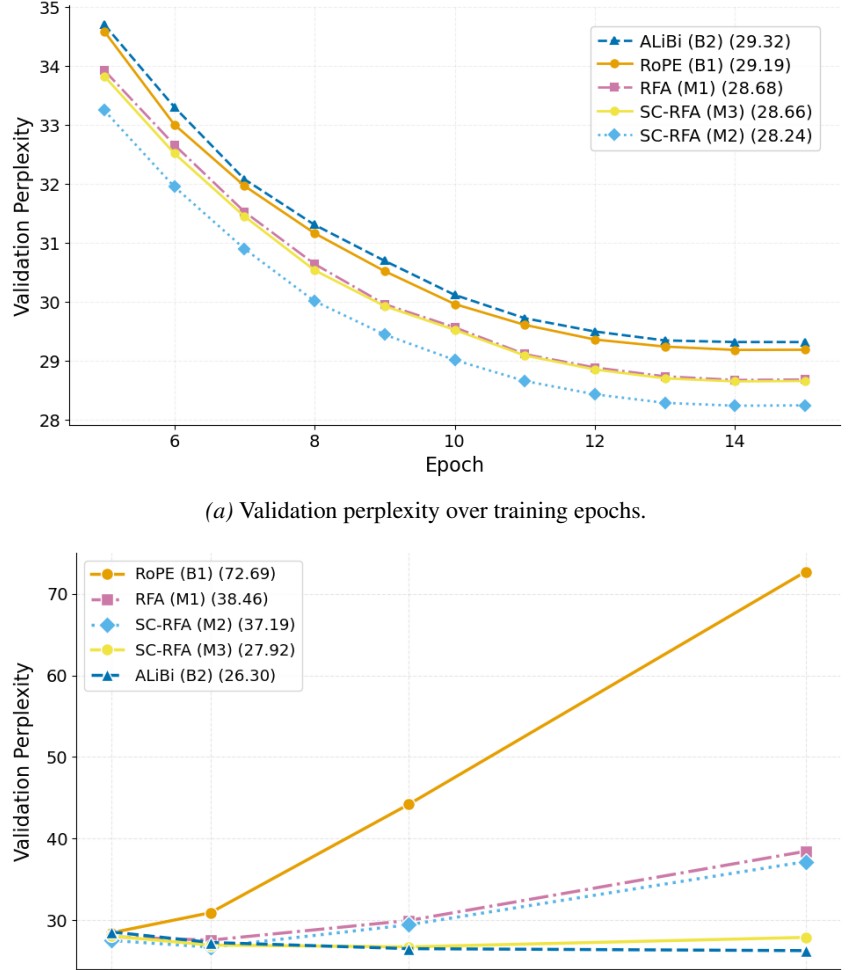

*(a)* Validation perplexity over training epochs.

*(b)* Test perplexity under length extrapolation beyond the training window (512 tokens).

*Figure 7.* **Training dynamics and length extrapolation on WikiText-103.** RFA variants converge faster during training and degrade more gradually with increasing context length than RoPE, while ALiBi remains stable due to enforced locality.

Figure 8 shows the sensitivity analysis over damping values $b$ reported in Table 2. Increasing damping improves long-range stability by suppressing high-frequency propagation, but excessively large damping degrades short-range modeling.

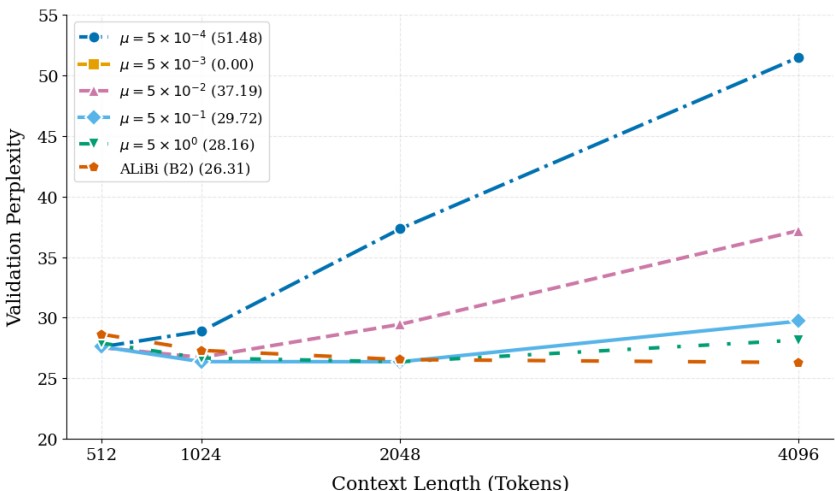

*Figure 8.* **Impact of the Damping Coefficient** $b$ **on Length Extrapolation.** Perplexity curves for varying $b$ demonstrate that higher damping coefficient values effectively stabilize long-range integration.

## F.2. Parameter Dynamics in RFA

Learned measurement and process noise parameters over the course of training are shown in Fig. 9 for the last layer of both RFA (M1) and SC-RFA (M2). Distinct trajectories in query and key noise parameters indicate that different heads self-organize into separate signal-to-noise regimes. We plot robustness parameter $\nu_s = \nu/d$ and inverse temperature $\beta_s$ in Fig. 10.

In general, lower-decay heads tend to converge to lower measurement noise variances $\eta^2, \gamma^2$, lower robustness parameter $\nu_s$, and higher inverse temperature $\beta_s$, consistent with stable long-range integration, while higher-decay heads tend to tolerate larger measurement noise.

Intermediate heads tend to converge to the highest measurement noise variance, lowest steady-state process uncertainty, and strongest robustness, consistent with modeling heterogeneous and noisy mid-range structure, while extreme short- and long-range heads tend to remain more tolerant to outliers.

The spectrally coupled model (M2, SC-RFA) exhibits lower average query and key noise variance and more clustered trajectories across heads.

When initialized in the diffusive regime, we observed that higher-decay heads consistently transitioned into the integrative regime ($\alpha > 0$), while the lowest-decay heads remained diffusive (Fig. 11).

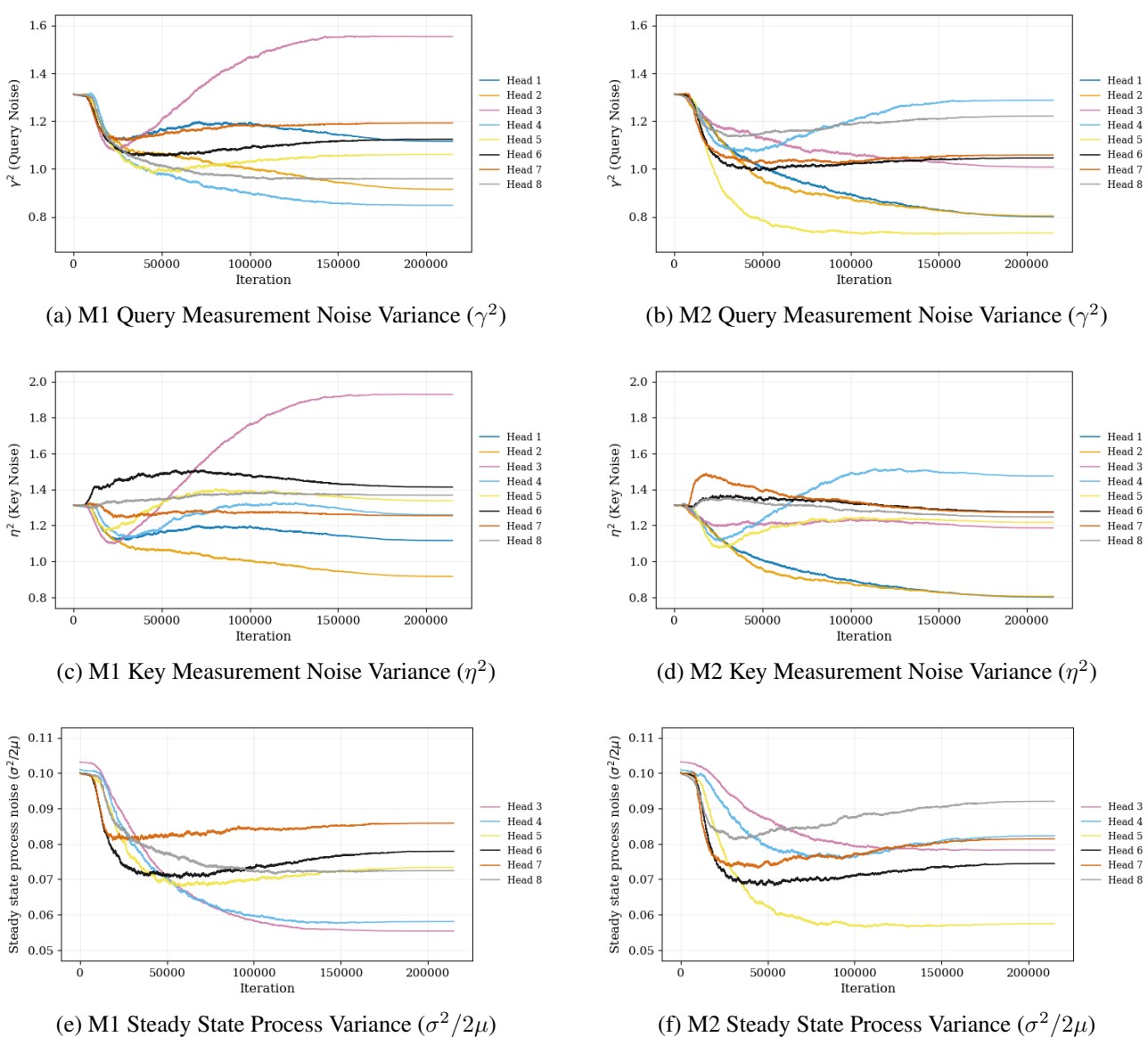

(a) M1 Query Measurement Noise Variance ($\gamma^2$)

(b) M2 Query Measurement Noise Variance ($\gamma^2$)

(c) M1 Key Measurement Noise Variance ($\eta^2$)

(d) M2 Key Measurement Noise Variance ($\eta^2$)

(e) M1 Steady State Process Variance ($\sigma^2/2\mu$)

(f) M2 Steady State Process Variance ($\sigma^2/2\mu$)

*Figure 9.* **Measurement and Process Noise Parameters Comparison.** Query and key measurement noise variance and state process variance for M1 and M2, over the course of training. (Note that $\tilde{\sigma}^2$ is undefined for heads 0 and 1, with $\mu = 0$.)

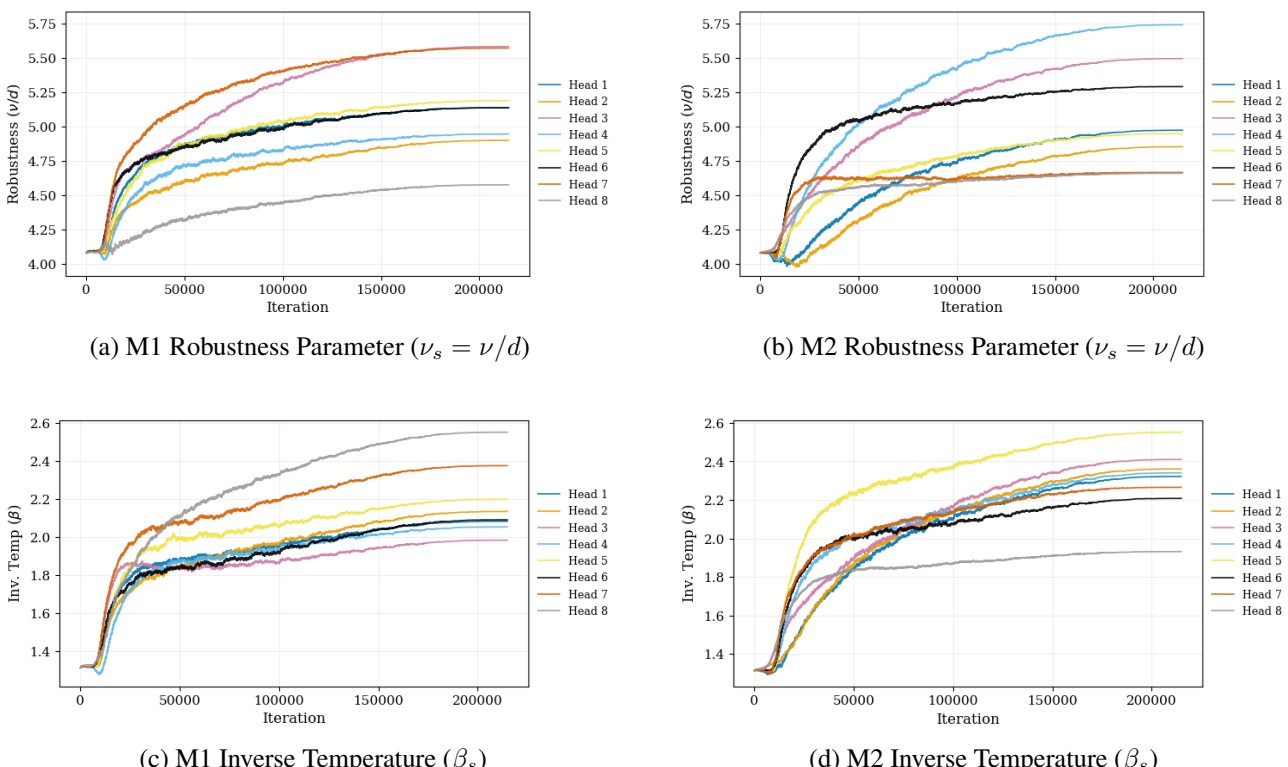

(a) M1 Robustness Parameter ($\nu_s = \nu/d$)

(b) M2 Robustness Parameter ($\nu_s = \nu/d$)

(c) M1 Inverse Temperature ($\beta_s$)

(d) M2 Inverse Temperature ($\beta_s$)

*Figure 10.* **Robustness and inverse temperature.** Robustness parameter and inverse temperature for M1 and M2, over the course of training.

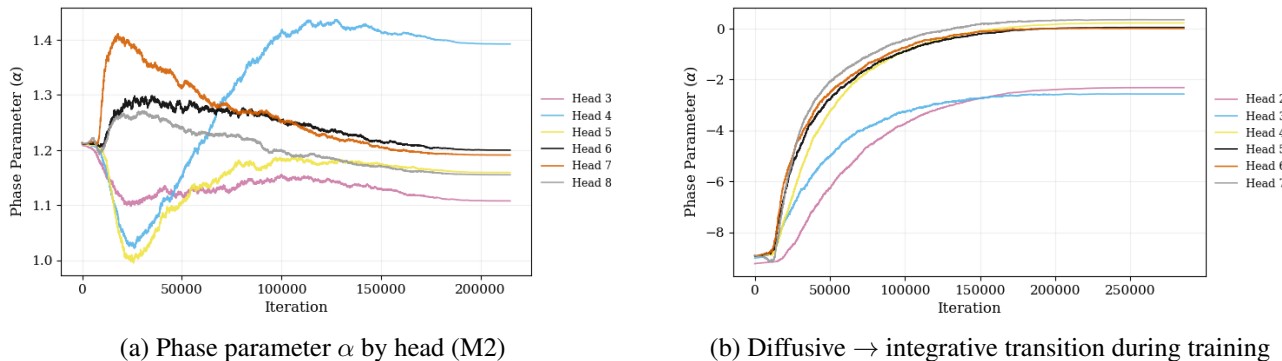

(a) Phase parameter $\alpha$ by head (M2)

(b) Diffusive $\rightarrow$ integrative transition during training

*Figure 11.* **Integrative dynamics in SC-RFA.** (a) Phase parameter ($\alpha$) under standard initialization, showing specialization across heads. (b) When initialized in the diffusive regime ($\alpha < 0$), most heads transitioned into the integrative regime ($\alpha > 0$) during training, while the two lowest-decay heads remained diffusive. (Note that $\alpha$ is undefined for heads 0 and 1, with $\mu = 0$.)

## F.3. Analysis of Attention Matrices

We plot attention matrices at a context length of 4096 to visualize long-range behaviors induced by each positional prior: the baselines RoPE (B1) (Fig 12) and ALiBi (B2) (Fig 13); and the RFA (M1) (Fig 14) and SC-RFA (M2) (Fig 15) models. We use attention matrices from the last layer of each model.

**Note:** For the RFA models, for improved visualization, we plot the unattenuated attention matrix $A$ rather than the decayed attention matrix $\hat{A} := A \odot E$.

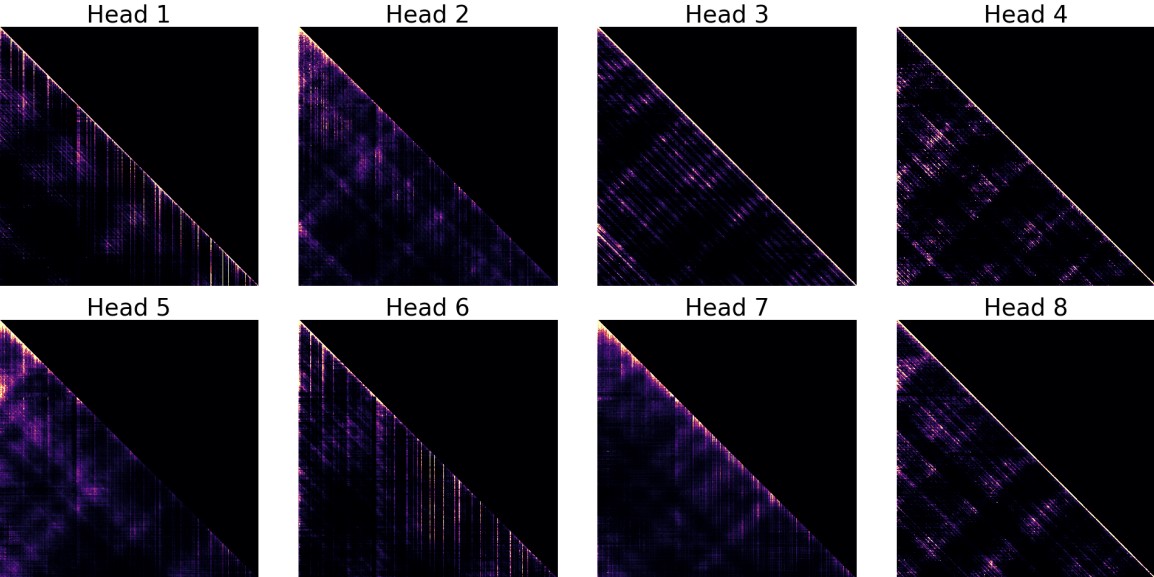

*Figure 12.* **Baseline RoPE Transformer (B1) at** $L = 4096$**:** Attention map exhibits persistent checkerboard structure.

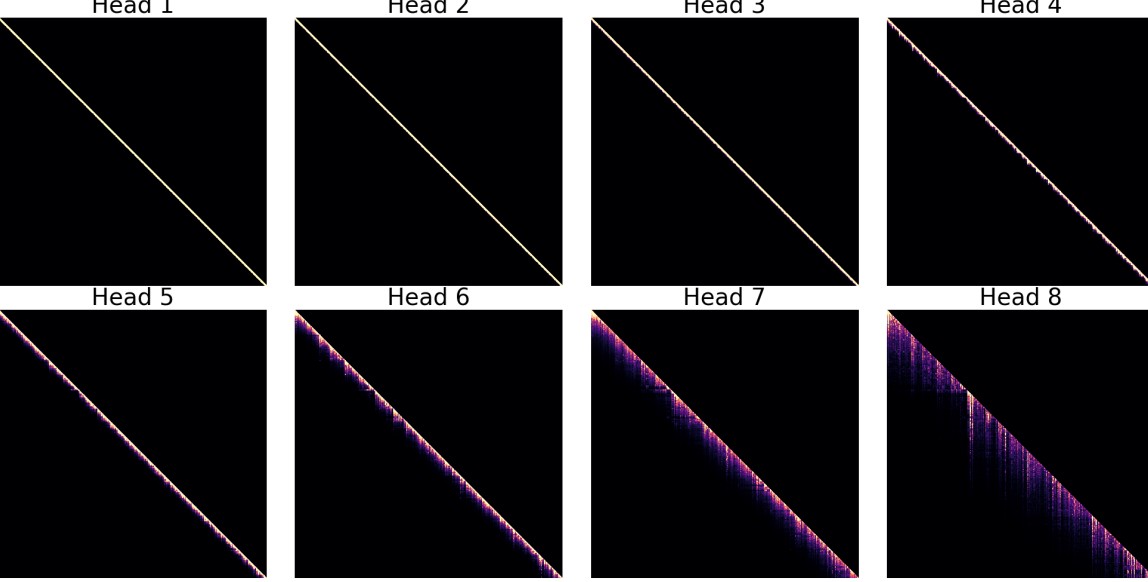

*Figure 13.* **ALiBi Transformer (B2) at** $L = 4096$**:** Attention maps remain tightly localized to the diagonal across all heads, with only modest widening in higher heads. Long-range structure is suppressed rather than integrated.

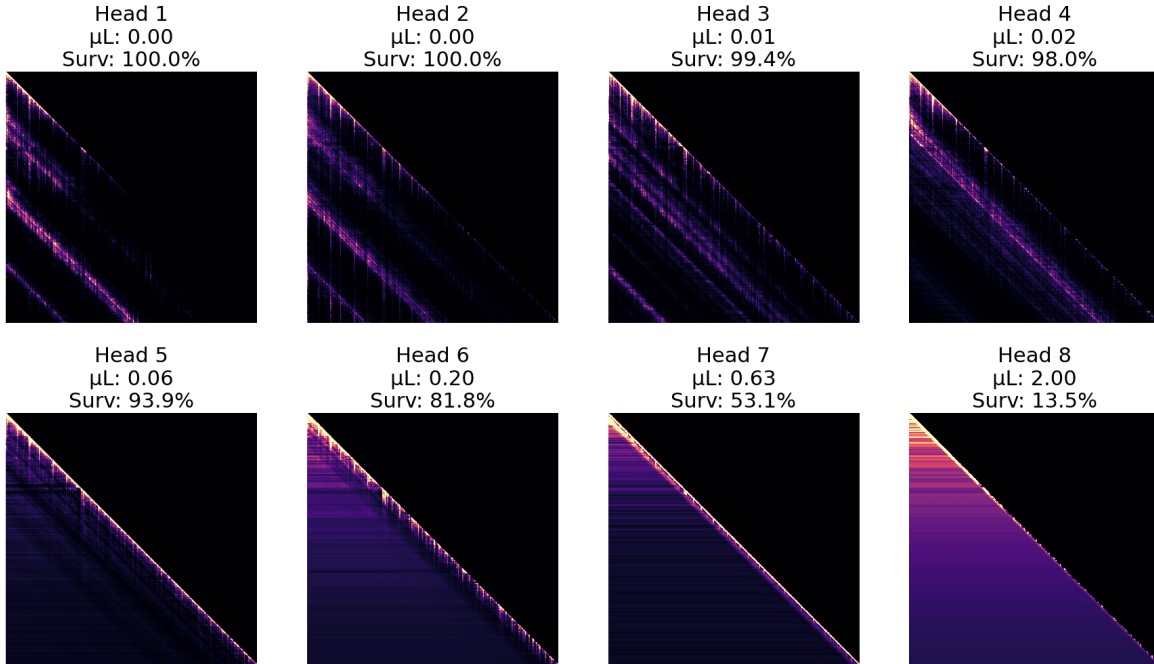

*Figure 14.* **Robust Filter Attention (M1) at** $L = 4096$**:** Periodic bands are clearly visible. High-decay heads concentrate focus on the local diagonal, while low-decay heads exhibit the integrative regime: the bottom-right corner near the diagonal is suppressed as the model waits for the SDE dynamics to suppress initial measurement noise before assigning high precision to the state estimate.

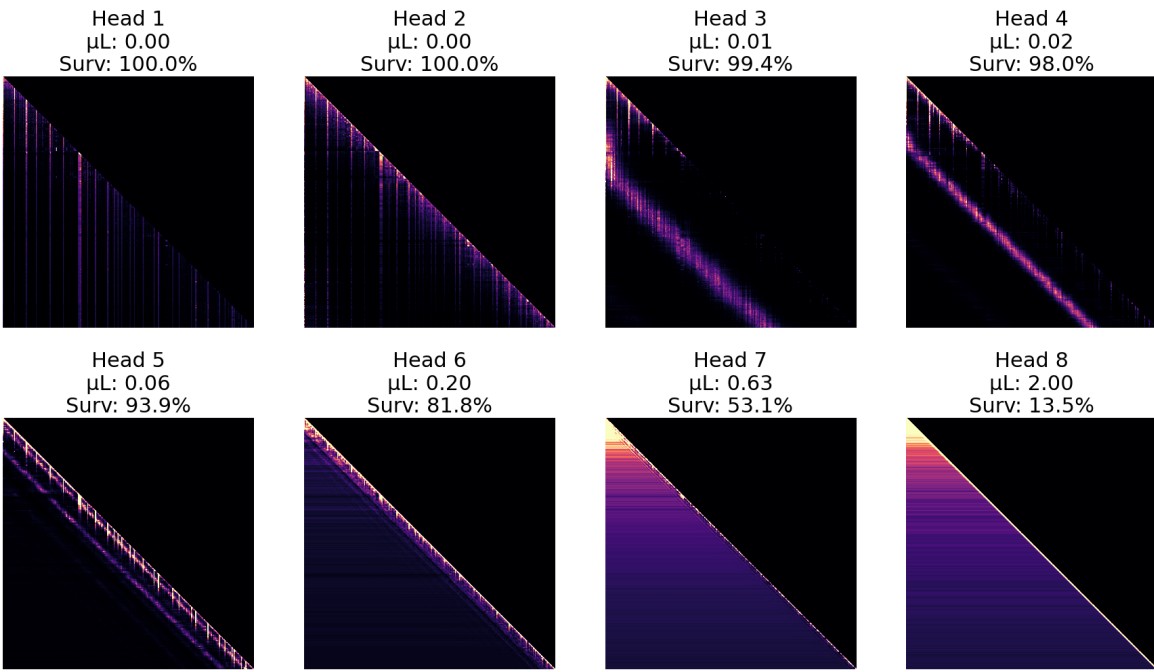

*Figure 15.* **Spectrally-Coupled RFA (M2,** $b = 0.05$**) at** $L = 4096$**.** Frequency-dependent damping ($\mu_h = b \cdot \omega_{h,\max}$) substantially alters long-range attention structure. SC-RFA has fewer periodic bands than RFA. Heads 3-5 each have only a single band, which become narrower and moves closer to the diagonal as decay increases. Heads 1 and 2 act as stable long-range integrators.

The attention maps for RoPE exhibit persistent checkerboard structure and high-frequency oscillations that remain visible even at large temporal offsets. In the absence of decay, these oscillations introduce non-local interference and unstable long-range patterns.

In ALiBi, attention remains tightly localized to the diagonal across all heads, reflecting its fixed distance-based bias. While higher heads show modestly broader receptive fields, long-range context is suppressed rather than integrated.

In contrast, RFA (M1) produces clearer periodic bands by aggregating in a stationary frame, preserving phase relationships and reducing interference. Some heads exhibit an "opening gate" behavior, where attention is suppressed near the diagonal and peaks at a characteristic lag, indicating delayed aggregation until the state estimate stabilizes.

Spectral coupling in SC-RFA (M2) sharpens and organizes this structure. Coupling decay to frequency ($\mu_h = b \cdot \omega_{h,\max}$) induces a specialization of heads to temporal lags: high-decay heads concentrate near the diagonal, while low-decay heads shift toward longer lags, forming distinct, narrow bands consistent with $\Delta t^* \propto 1/\mu_h$ (Sec. 3.7).

In SC-RFA, the first two (zero-decay) heads exhibit vertical structures corresponding to stable long-range retrieval. These heads effectively perform global key–value lookup rather than temporal filtering, assigning similar weight to salient tokens across all query positions and producing clean, vertically aligned patterns.

Together, these patterns support the view that RFA learns a structured multi-scale filtering behavior rather than relying on fixed geometric positional biases.

