# OpenReview forum: "Robust Filter Attention: Self-Attention as Precision-Weighted State Estimation"
_ICML.cc/2026/Conference — ICML 2026 spotlight_

### Official Review · Reviewer_xYZW · 2026-02-27

**Soundness:** 3
**Presentation:** 3
**Significance:** 2
**Originality:** 3
**Overall Recommendation:** 5
**Confidence:** 5

**Summary:**

This work presents Robust Filter Attention (RFA), an adaptation of the conventional Transformer attention mechanism aimed at enhancing resilience and frequency selectivity in sequence modeling. The fundamental concept is to reconceptualize attention as a filtering mechanism in the spectral domain and to deliberately regulate its frequency response via structured filter design. The authors conceptualize attention not merely as a similarity-based weighting mechanism, but as a learnable filter operator capable of attenuating extraneous high-frequency noise while maintaining semantically significant signal components.

The proposed method incorporates a filtering mechanism into the attention calculation, resulting in enhanced stability against perturbations and superior generalization in noisy or distribution-shifted environments. The work presents a theoretical examination of the induced operator, encompassing its frequency characteristics and stability qualities, while contrasting it with conventional softmax attention.

The method is empirically assessed using synthetic robustness benchmarks, long-range sequence tasks, and language modeling studies. The results indicate enhanced robustness to input corruption and competitive or superior performance compared to baseline Transformer topologies, especially in environments with noise or adversarial perturbations.

The study offers a filter-theoretic viewpoint on attention and assesses its consequences for resilience and stability in Transformer models.

**Compliance With Llm Reviewing Policy:**

Affirmed.

**Final Justification:**

The author has responded with a constructive and professional attitude, and the issues raised in the previous review have been adequately addressed. The revisions have satisfactorily resolved my concerns. Congratulations!

**Key Questions For Authors:**

1. Validation of robustness mechanisms.
The research contends that the filtering concept explicitly enhances robustness by modifying the frequency response of attention. Could the authors furnish more robust scientific evidence delineating this mechanism (e.g., ablation studies directly contrasting learnt frequency responses or controlled perturbation experiments)? A more explicit causal relationship between the theoretical filter features and the observed robustness improvements would bolster the assertions and potentially enhance my evaluation.

2. Generalization to extensive models.
The experiments are performed in small to medium-scale environments. Do the authors possess preliminary proof that the proposed adjustment preserves its advantages in large-scale language models? If enhancements in robustness continue at scale without compromising clean performance, this would greatly amplify the significance of the effort.

3. Evaluation of alternative robustness techniques.
How does Robust Filter Attention compare to alternative robustness-focused interventions (e.g., adversarial training, regularization methods, attention smoothing techniques) under equal computational budgets? A more explicit comparison might enhance the precision of the contribution's positioning.

4. Compromises between resilience and expressiveness.
Does the implemented filtering framework limit the expressive capability of attention under pristine, noise-free conditions? An examination of potential trade-offs, such as in long-range semantic accumulation tasks, would elucidate if the enhancements in resilience incur any representational costs.

**Limitations:**

The authors address constraints mainly with the experimental scale and the extent of robustness assessment. The study concentrates on controlled noise and perturbation environments, although larger real-world robustness scenarios, such as large-scale deployment contexts or diverse multimodal inputs, are unexamined. The theoretical study offers insight into frequency behavior but fails to provide comprehensive end-to-end assurances for trained models.

The work seemingly does not present any direct negative societal hazards beyond those intrinsic to massive language or sequence models. Nevertheless, as to other architectural alterations, downstream exploitation is feasible if implemented in high-stakes settings without adequate safeguards.

**Strengths And Weaknesses:**

Strengths：
1. Distinct conceptual redefinition of attention.
The research offers a reasoned reframing of attention as a structured filtering operator instead of merely a similarity-based weighting mechanism. This filter-theoretic perspective is logically consistent and provides an alternate framework for comprehending robustness and stability in Transformers.

2. Technical foundation and analytical assistance.
The authors enhance the architectural approach with a theoretical examination of the induced operator, encompassing frequency-domain interpretations and stability issues. Although not entirely comprehensive, the research offers significant insight into the reasons the suggested technique may mitigate noise or disturbances.

3. Emphasize resilience.
The research specifically addresses robustness in the presence of noise and disturbances, a matter of growing significance in practical applications. The empirical part encompasses corruption-style assessments and stress testing that exceed conventional clean norms.

4. Empirical Competitiveness.
Robust Filter Attention demonstrates competitive or enhanced performance compared to typical Transformer baselines across several tasks, especially in scenarios characterized by noise or distribution shifts.

Weaknesses：

1. Extent of theoretical assurances.
Although the filtering interpretation is attractive, the formal assurances seem constrained in extent. The relationship between frequency-domain analysis and the behavior of end-to-end learning models could be more thoroughly substantiated.

2. Empirical scope.
Despite the inclusion of robustness benchmarks, the evaluation is still limited in terms of model scale and task diversity. The persistence of benefits in extensive language models or across diverse domains is uncertain.

3. Comparison with roughly related robustness methodologies.
The manuscript would improve with more explicit comparisons to other robustness-focused Transformer changes or regularization techniques, to more effectively delineate the impact of the suggested filtering process.

4. Precision in placing.
The connection between Robust Filter Attention and current spectral or filtering interpretations of attention should be defined more clearly to enhance assertions of originality.

---

> ### Author Rebuttal · Authors · 2026-03-30
>
> We thank the reviewer for their feedback. However, we believe there may be a mismatch in interpretation between parts of the review and the content of the paper. The core contribution of our work is a model-based derivation of attention as an approximate precision-weighted estimator under a dynamical prior, rather than a robustness or frequency-domain method. We appreciate the reviewer’s perspective and welcome the opportunity to clarify the intended scope of the work.
>
> The review describes the method as operating in the “spectral domain,” regulating “frequency response,” and analyzing “frequency characteristics of the induced operator.” The reviewer's intuition that RFA implements a form of structured filtering is correct. However, our filtering occurs in the time domain through dynamical state propagation, not through spectral decomposition of the attention operator. The frequency–decay coupling in SC-RFA is introduced for stability (e.g., to avoid aliasing under discretization), rather than to explicitly shape the frequency response of the attention operator.
>
> The review refers to evaluation settings that are not present in our work, including “synthetic robustness benchmarks,” “corruption-style assessments,” “input corruption,” and experiments under “noise or adversarial perturbations” or “distribution shifts.” Our paper does not include robustness benchmarks, adversarial perturbations, or corruption-based evaluations. The empirical study focuses on language modeling and long-context extrapolation (WikiText-103 and BabyLM-2025), as described in Section 4.
> The review frames the work as addressing “robustness to noise and disturbances” as a primary objective. We clarify that the term “robust” in Robust Filter Attention refers to robustness in the M-estimation sense, not adversarial or distributional robustness. Specifically, $w_{ij}$​ arises from standard M-estimation as a data-dependent influence function that down-weights observations inconsistent with the dynamical prior. This reflects a likelihood choice (e.g., heavy-tailed vs Gaussian), not a separate robustness objective or training procedure.
>
> The reviewer raises concerns about empirical scope and scaling. We note that the experiments are conducted on standard language modeling benchmarks with controlled comparisons to RoPE and ALiBi, and include an extensive set of ablations (20 trained models in total).
>
> The reviewer suggests comparisons to adversarial training and other robustness-focused techniques. We note that RFA is not a robustness intervention or training strategy, but a reformulation of the attention mechanism itself. Accordingly, the most relevant comparisons are to alternative positional or attention formulations (e.g., RoPE and ALiBi), which we include.
>
> The reviewer suggests experiments involving frequency response analysis or perturbation-based robustness evaluation. These directions are orthogonal to the current paper’s focus. Our empirical evaluation instead targets long-context extrapolation and temporal consistency, which directly follow from the proposed dynamical formulation.
>
> The reviewer asks whether the proposed filtering mechanism reduces expressiveness. As we show in the text, RFA is a strict generalization of standard attention: in the zero-noise, zero-decay limit, it recovers RoPE exactly. Therefore, it does not restrict expressiveness, but rather introduces a learned, uncertainty-aware generalization of existing positional mechanisms.

---

> > ### Author Rebuttal · Reviewer_xYZW · 2026-03-31
> >
> > The authors effectively elucidate that the primary contribution is a dynamical-prior-based reconfiguration of attention as an approximate precision-weighted estimator, and that "robust" is employed in the context of M-estimation rather than adversarial or distributional robustness. The empirical evaluation focuses on language modeling and long-context extrapolation, rather than benchmarks based on corruption robustness. These answers alleviate my primary worries, prompting me to revise my opinion more favorably.

---

### Official Review · Reviewer_6ud1 · 2026-03-09

**Soundness:** 2
**Presentation:** 3
**Significance:** 3
**Originality:** 3
**Overall Recommendation:** 4
**Confidence:** 3

**Summary:**

This paper introduces Robust Filter Attention (RFA), an attention mechanism that views attention as a filtering process over time under uncertainty. The method builds on rotary positional embeddings and aims to improve long-context stability. The authors also propose an extended version called Spectrally-Coupled RFA (SC-RFA), which couples rotation frequencies with decay rates across attention heads. The proposed methods are evaluated on two language modeling datasets, WikiText-103 and BabyLM-2025, and compared with two baselines, RoPE and ALiBi. The results show that the proposed methods consistently outperform RoPE and achieve better performance than ALiBi at shorter extrapolation lengths (up to 1024 tokens). The authors also conduct sensitivity analysis and ablation studies to examine the contribution of different components of the method.

**Compliance With Llm Reviewing Policy:**

Affirmed.

**Final Justification:**

I thank the authors for their response. The expanded theoretical explanation has improved my understanding of the work, and based on this, I increase my score to weak accept.

However, the empirical evidence remains moderate and does not fully support the claims, particularly for long-context performance, where strong baselines such as ALiBi remain competitive or superior. The discussion of the trade-off controlled by the damping parameter mainly explains the mechanism, but practical guidance remains unclear, including when the method should be preferred and how this parameter should be selected.

**Key Questions For Authors:**

Q1. The experiments evaluate long-context extrapolation up to 4096 tokens. Could the authors report how the method performs for longer contexts (e.g., 8k or 16k tokens)?

Q2. Since the paper discusses time and memory complexity, could the authors also provide empirical measurements of training time, inference time, and training memory usage?

Q3. In the introduction, the authors mention that “in the zero-noise and zero-decay limit, the formulation reduces to a purely rotational embedding consistent with RoPE.” Could the authors clarify how this relationship is reflected in the experimental results shown in Table 1?

Q4. The models are trained for 15 epochs. It would be helpful to clarify whether the models have fully converged, or to provide training curves to show that performance has stabilized.

Q5. Section 4.4 states that “RFA variants converge faster and achieve lower validation perplexity earlier in training than RoPE and ALiBi”. However, Figure 6(a) seems to show similar convergence trends across all methods. It would be helpful to clarify this observation.

**Limitations:**

Yes

**Strengths And Weaknesses:**

### Strengths

1. The paper addresses an important problem in improving attention mechanisms and long-context modeling in Transformer architectures.

2. The paper includes a detailed theoretical formulation with comprehensive derivations and analysis, which strengthens the technical rigor of the work.

3. The authors perform ablation studies to analyze the contribution of different components of the method.

---

### Weaknesses

1. While the proposed method improves over RoPE, ALiBi still achieves substantially lower perplexity at the longest tested contexts, which raises questions about the claimed improvement in long-context stability.

2. While the related work section discusses existing literature, the introduction mainly motivates the method through comparison with RoPE and ALiBi. Expanding the introduction to more clearly position the proposed method with respect to a broader range of existing attention or positional encoding methods would help clarify its novelty.

3. Although the paper reports the time and memory complexity of the proposed method, it is not clearly explained how these complexities are derived.

4. Experiments are conducted on relatively small Transformer models (6 layers, d=256). Evaluating the proposed mechanism on larger models could help demonstrate its scalability and robustness.

---

> ### Author Rebuttal · Authors · 2026-03-30
>
> We thank the reviewer for their constructive feedback.
>
> As we note in our response to Reviewer 3G3G, the gap between SC-RFA and ALiBi at long contexts reflects a principled trade-off controlled by the damping parameter b, not a fundamental limitation of the method. ALiBi achieves long-range stability by suppressing all distant interactions, which is equivalent to setting precision to zero beyond a certain lag. SC-RFA instead maintains nonzero but attenuated precision at long range, which costs some stability but improves performance at intermediate range. Table 2 shows this trade-off is continuously controlled. M2.5 ablates complex rotations from RFA, effectively matching the performance of ALiBi at the cost of lower accuracy in the near-field.
>
> We have rewritten the Introduction to more clearly indicate the main ideas. We note that the Related Work section compares RFA to (a) Probabilistic and Kernel Views of Attention; (b) Filtering, Continuous Dynamics, and SSMs; and (c) Positional Encodings and Complex Geometry. We have partially rewritten the latter subsection to help clarify the connection to existing methods.
>
> The primary claims of this paper are theoretical — that attention can be derived as an approximate precision-weighted estimator under a linear SDE, and that this derivation recovers existing methods as limiting cases. These claims do not depend on model scale. The empirical evaluation is designed to isolate the effect of the positional mechanism specifically, which is most cleanly done at smaller scale where other factors (optimizer dynamics, data scale, architectural choices) are controlled. The use of (≈20M parameter) models enabled an extensive ablation study (20 models in total). We agree that evaluating at larger scale is an important direction for future work and will note this explicitly as a limitation of the current work.
>
> RFA involves the same matrix and vector operations as ordinary attention, therefore the asymptotic complexity is the same. The $O(N^2 + Nd)$ memory complexity follows from the factorization derived in Appendix B, Section B.2. The key step is that the isotropic constraint causes the precision kernel to depend only on the scalar time lag $|i-j|$, not on the feature index $k$. This allows it to factor outside the feature summation, so the cross-term $Re(\tilde{Q}^\dagger \tilde{K})$ can be computed as a single $O(N^2 d)$ matrix multiplication without storing $O(N^2 d)$ intermediate tensors. The additional terms $||Q_i||^2$ and $E[i,j]^2 ||K_j||^2$ are computed via broadcasting at $O(Nd)$ cost. We will add a concise summary of this derivation to the main text.
>
> Regarding evaluation at 8k or 16k tokens: the causal mask in our implementation was set to a maximum of 4096 tokens for memory reasons, as memory requirements scale quadratically with context length at standard attention complexity. The long-context experiments at 4096 tokens is 8 times the training window, at which point RoPE models have exploded. Extending to 8k or 16k would require either a larger compute budget or a sparse attention approximation, both of which are outside the scope of the current work. We will note this explicitly.
>
> RFA preserves the same operations as standard attention with the addition of elementwise operations for the precision kernel and decay factors, which are $O(N^2)$ and $O(Nd)$ respectively and negligible relative to the dominant $O(N^2 d)$ matrix multiplication. We will clarify this. As we note in Appendix D.3, “Our current implementation is written in high-level PyTorch and incurs an approximately 2x training overhead relative to PyTorch’s optimized scaled dot-product attention backend.” The inference overhead is similar. However, RFA is compatible with FlashAttention and we expect that future kernel fusion would reduce this overhead to be negligible.
>
> In the zero-noise, zero-decay limit ($\mu=0, \sigma^2=0$), the precision kernel becomes constant and the residual reduces to a scaled dot product between rotated queries and keys, as in RoPE. Table 1 shows a ladder of ablations which progressively ablate out portions of RFA until we arrive at M2.6, which is equivalent to RoPE with value rotation/counter-rotation. Ablating out value rotation/counter-rotation from M2.6 would leave us with exactly RoPE, hence we do not include a separate ablation.
>
> Yes, the models are trained until convergence. We will clarify this in the text.
>
> The reviewer correctly points out that our claim in Section 4.4 that RFA variants “converge faster and achieve lower validation perplexity earlier in training” is not accurate. What this figure shows is that RFA variants “achieve lower validation perplexity earlier in training,” and that this is consistent throughout training. This was the point of this figure. We have removed the language about “converging faster” which was a sloppy misuse of that term on our part.

---

> > ### Author Rebuttal · Reviewer_6ud1 · 2026-04-02
> >
> > Thank you for the response and for providing additional details, especially regarding time and memory complexities.
> >
> > While I appreciate the strong theoretical formulation, the empirical results are still not fully convincing. The proposed method introduces approximately 2× training overhead, while showing weaker performance than ALiBi at longer contexts. The improvement appears to mainly benefit intermediate context lengths. In modern LLM settings, where long-context stability is often important, methods with ALiBi-like behavior may still be preferable.
> >
> > Regarding the ''principled trade-off controlled by the damping parameter'', it is not clear how this trade-off should be evaluated in practice. What defines a good trade-off? From Table 2, it is also unclear under which damping coefficient the proposed method can outperform ALiBi in long-context settings.
> >
> > Also, it remains unclear whether training for 15 epochs is sufficient for full convergence. In the original papers of the baselines used in this work, longer training is used (e.g., 200+ epochs). Even if the same setting is applied to baselines, it is difficult to rule out whether some of the observed performance differences are due to undertraining.

---

> > > ### Author Response · Authors · 2026-04-08
> > >
> > > We thank the reviewer for the additional feedback.
> > >
> > > (1) Training overhead
> > >
> > > RFA preserves the same computational structure as standard attention: the dominant operation remains a single $\mathcal{O}(N^2 d)$ matrix multiplication. The additional components (decay and precision kernels) are elementwise operations with $\mathcal{O}(N^2)$ and $\mathcal{O}(Nd)$ cost, respectively, and do not change asymptotic complexity. Because of this, RFA is fully compatible with fused attention kernels (e.g., FlashAttention), and we expect the overhead to be negligible in optimized implementations.
> > >
> > > (2) Training Duration
> > >
> > > We used a cosine annealing schedule, under which both training and validation losses converged smoothly. Our models (20M parameters) were trained on $\sim$1.5B tokens ($\sim$75 tokens per parameter), which is comparable to or higher than typical scaling ratios at this size.
> > >
> > > The relative performance differences between methods emerge early and remain stable throughout training (Fig. 6A, Appendix F), and remain consistent across many ablations and training runs, which is evidence that the observed behavior is architectural rather than a result of training dynamics.
> > >
> > > (3) What defines a “good trade-off”?
> > >
> > > In RoPE, tokens separated by multiples of the rotation period map to nearly identical representations. As a result, at long context lengths, the model cannot distinguish between distant tokens whose phases have “wrapped around,” leading to spurious high-similarity interactions. While the model can learn to partially compensate for this aliasing problem within the training window, it becomes a major problem outside the training window.
> > >
> > > Decay helps to alleviate this problem by suppressing the magnitude of older cycles, so even if phases align after wrapping, their contribution is negligible. Using $\mu_h=b \cdot \omega_{h,max}$, the decay over one period is bounded by $e^{-b}$. With $b\ll1$, oscillatory structure persists across multiple cycles, which helps the model to resolve fine detail at large lags within the training window. When $b\gg1$, high-frequency modes decay within a single cycle, preventing phase wrap-around from producing spurious matches.
> > >
> > > As shown in Table 3, $b=0.05$ achieves the best training-window ($L=512$) performance of 27.54 PPL. At $b=5$, the model achieves a slightly worse 27.91 PPL at $L=512$, but nearly eliminates long-context degradation.
> > >
> > > RFA improves both in-window performance and extrapolation relative to RoPE. Unlike RoPE, which degrades monotonically beyond the training window, RFA maintains stable performance and even improves at intermediate distances before gradually degrading, which is practically valuable.
> > >
> > > (4) Why both decay and precision are necessary
> > >
> > > Figure 14 (Appendix F.3) shows that attention heads self-organize into a filter bank of time-localized bands through the interaction of decay and precision. In the integrative regime, precision increases with lag; combined with exponential decay, this produces a peak at a nonzero lag $\Delta t^*$, causing each head to specialize to a particular lag. High-frequency heads form bands near the diagonal, while slower heads specialize further out.
> > >
> > > When $\mu$ is small, $\Delta t^*$ is pushed beyond the observed sequence, so the model never observes the peak. In this regime, the integrative regime becomes ineffective and tends to accumulate noise. A diffusive profile is therefore preferable: monotonic decay suppresses distant low-SNR contributions and imposes a stable recency bias, improving extrapolation.
> > >
> > > We will discuss this in the main paper.
> > >
> > > (5) On ALiBi vs. SC-RFA at very long context
> > >
> > > We acknowledge that ALiBi remains the strongest baseline for extreme extrapolation. ALiBi achieves long-context stability by enforcing strict locality and discarding the high-frequency components responsible for RoPE’s fine-grained localization, eliminating phase ambiguity at the cost of reduced resolution. SC-RFA instead attenuates these frequencies as a function of distance, preserving local structure while suppressing long-range aliasing, and thus occupies a different point on the capability–stability tradeoff.
> > >
> > > In Table 1, the lowest-frequency heads were configured with $\mu=0$, $\tilde{\sigma}=0$ to preserve RoPE-like infinite-integration heads. For a better comparison with ALiBi, we trained a new model that sets all heads using the rule $\mu_h = b \cdot \omega_{h,max}$, with heavy damping ($b=5.0$), and higher initialization of $\tilde{\sigma}$.
> > >
> > > This new model achieves 28.11 PPL at $L=512$ and 27.86 PPL at $L=4096$, compared to ALiBi’s 28.64 PPL at $L=512$ and 26.31 PPL at $L=4096$. SC-RFA maintains a clear advantage within the training window while narrowing the long-context gap to 1.55 PPL.
> > >
> > > We will include these new results and analysis in the text.

---

### Official Review · Reviewer_Pzpd · 2026-03-12

**Soundness:** 3
**Presentation:** 2
**Significance:** 4
**Originality:** 4
**Overall Recommendation:** 5
**Confidence:** 3

**Summary:**

The authors present a new perspective on how to model temporal structure in a transformer, replacing positional embeddings with a prior over how past embeddings (keys) inform the current (query), in the form of a stochastic linear model of latent-state evolution. This prior is based on the assumption that the embeddings are observations emanating from a linear dynamical system. It seems as if this prior will yield recency-biased embeddings, in contrast to RoPE (which is biased toward cyclical attention patterns). RoPE and ALiBi are claimed to be limiting cases of the proposed method, RFA. An extension of RFA, SC-RFA is also presented which allows some attention heads to act as sharp short-range filters and others to act as smooth long-range integrators. Empirical evaluations show the proposed method shows promise.

**Compliance With Llm Reviewing Policy:**

Affirmed.

**Final Justification:**

I found insight and novelty in this work. I expect it will inspire others.

**Key Questions For Authors:**

What justifies treating the generative process as a line time-invariant dynamical system?

The matrix C in eqn 1 is assumed to be invertible. How can this possibly be given that x is going to be of much higher dimensionality than z?

I'm wondering if all this fancy math just boils down to a relatively simple type of recency based prior on attention. Can you characterize the form of the decay with distance? Is it exponential or something?

If I've grokked the main idea, might there be a connection to the recent literature on considering state sequences to be produced by a (nonlinear) dynamical system e.g., https://arxiv.org/abs/2511.05963, https://arxiv.org/abs/2602.22617

Eqn 2 mentions Q, which must be related to the Wiener process but it's not defined.

What justifies Wiener process noise or Gaussian observation noise?

lines 165-166: What is the motivation/justification for reweighting by the prior precisions?  What are the "desired properties of estimator"?

**Limitations:**

Yes

**Strengths And Weaknesses:**

The greatest strength of the work is that it develops a very original idea. To be honest, I didn't really care about the evaluations because I felt like there was so much to learn in understanding the basic framework, which seems like a principled way of considering the influence of contextual information on the current input.

The greatest weakness of the work is that it makes a boatload of assumptions which do not seem terribly realistic but are needed for the elegant mathematical framework and to make the approach tractable. I didn't buy most of the assumptions, but the framework itself seems very interesting.

Most of the paper was way over my head, but I hope to have gleaned a high level understanding of the approach. While the paper is competently written, the introduction is unfortunately focused on emphasizing contributions and giving a succinct summary rather than making the work more broadly accessible. I couldn't get a gist of what was going on until section 3.1. I also wasn't able to follow most of Section 3; my understanding doesn't go beyond the fact that a bunch of additional assumptions are made to reduce the parameters of the linear dynamical system to a small number of scalars, which can be learned.

---

> ### Author Rebuttal · Authors · 2026-03-30
>
> We would like to thank the reviewer for their comments, which have been helpful in clarifying the method.
>
> The assumptions used by RFA are not assumptions on the generative model that actually produced the data. Rather, these are assumptions on the form of the propagation used when comparing keys to queries. A linear SDE is used for the same reason it is used in the Kalman Filter, which is that it allows for closed-form propagation of the mean and covariance. Closed-form propagation is also the reason a linear model is used in RoPE, though the latter lacks a notion of uncertainty. Furthermore, since RFA is a strict generalization of dot product attention and RoPE, its assumptions are at least as general as these other attention mechanisms.
>
> We have substantially revised the Introduction and Methods sections to improve clarity and provide more intuition. The Appendix shows that attention is a degenerate case of a tensor operation, which is, however, intractable, due to the $O(N^2 d)$ memory requirement. As shown in the Appendix, one cannot have an RFA mechanism that is simultaneously tractable, stable, and has anisotropic decay/noise. The assumption of isotropic decay and noise is thus forced upon us in order to maintain tractability and stability. However, the isotropic constraint still allows heads to specialize into qualitatively distinct filtering regimes (see Appendix F.2).
>
> We chose an invertible $C$ because it is natural for attention, it is the simplest case, and it already gives us a strict generalization of prior methods. In RFA, we treat keys and queries as noisy observations of the same latent process. Since the keys and queries live in the same embedding dimension, $C$ is square. If $C$ were not full rank, multiple latent states would produce identical observations, making the estimation problem ill-posed without additional constraints. Examining the case of non-invertible $C$ would be interesting but is outside the scope of the current work. We will address this issue in the text.
>
> Standard attention is a kernel smoother where the kernel is chosen by learning. RoPE adds structure corresponding to a zero-decay, zero-noise LTI system to that kernel. What RFA does differently is enforce dynamical consistency between two things that in all prior work are independent design choices: how you transport representations across time, and how you weight them. RoPE defines transport without any covariance model; ALiBi imposes a distance penalty without a transport model. In RFA these are not two separate choices — they're both consequences of the same underlying SDE. This was not clear enough in the original draft. We have rewritten the Introduction to clarify this point.
>
> RFA is not just a recency-based prior, because (a) it is not a heuristic prior, but rather one that is consistent with the linear dynamics (in particular, with the decay $\mu$), (b) it allows for two regimes: a diffusive regime (a recency prior), but also an integrative regime, in which more distant keys are actually more informative, because stable dynamics dissipate noise, (c) the same kernel $P_{\Delta t}$ enters both additively and multiplicatively (see Appendix B.3.3), and (d) it shows that value rotation/counter-rotation are essential for dynamical consistency. Removing value rotation causes catastrophic failure at long context (PPL 463 at L=4096), which no recency bias explanation can account for, since a decay kernel on attention scores does not affect the value stream.
>
> There is some connection to the papers you cite, though they are different approaches. NextLat modifies the training objective to encourage dynamically-consistent representations without changing the attention mechanism. RFA modifies the attention mechanism to encode a dynamical prior without changing the training objective. The STP paper argues that well-trained LLMs should have approximately linear hidden state dynamics. RFA explicitly models the hidden state dynamics as linear by construction, through the linear SDE.
>
> You are correct that $Q$ is not defined in the main text. We have added the definition ($Q = G G^T$), and other missing definitions, to the text.
>
> Wiener process noise and Gaussian measurement noise are used for the same reason they are used in the Kalman Filter, which is that they allow for closed-form propagation of the mean and covariance. We discuss an extension to colored noise in Appendix C.
>
> The reason we reweight by the precisions is because this is the optimal solution (under the conditional independence approximation, without which the solution would not admit a tractable parallel form). We agree that the reference to “desired properties of the estimator” was unclear. The intended point was that the specific form of $w_{ij}$ is not constrained by the formulation, but can be chosen by the user, including the forms we provide but also allowing others. We will clarify this and the other points you have raised in the text.

---

> > ### Author Rebuttal · Reviewer_Pzpd · 2026-03-31
> >
> > I thank the authors for their detailed and helpful responses to my questions. I already gave this paper a rating on the high side. Whatever weaknesses the work has, the ideas are sufficiently interesting and mature that the larger community would benefit by hearing them. (I already have 2 colleagues I'm itching to show the paper to, so please slap on arXiv if the conference isn't able to accept this version!)

---

### Official Review · Reviewer_3G3G · 2026-03-13

**Soundness:** 2
**Presentation:** 1
**Significance:** 3
**Originality:** 3
**Overall Recommendation:** 5
**Confidence:** 4

**Summary:**

This paper proposes to reformulate self-attention as a robust filtering problem, with assumptions and approximations allowing for preserving the same computational complexity as standard attention mechanisms. The performance is assessed against baselines for two datasets, Wikitext-103 and BabyLM-2025.

**Compliance With Llm Reviewing Policy:**

Affirmed.

**Final Justification:**

My main concerns were adequately addressed by the rebuttal.

**Key Questions For Authors:**

1. Several approximations and heuristics are used, what are these exactly and how much do they affect the obtain results? The assumptions are mentioned in the conclusion but not the approximations (in how the observations are taken into account) and heuristics such as the weighting of the precision matrix.
2. Table 2 seems to indicate that $b=5$ allows to improve on the performance of ALiBi for all values of $L$, although with a more modest improvement for small context lengths. Why not presenting these results in Table 1? Would the results in Table 3 degrade under this value of $b$?

**Limitations:**

There is no proper discussion around the limitations of the proposed method.

**Strengths And Weaknesses:**

# Strengths
1. The interpretation of attention as a robust filter is interesting.
2. The fact that the complexity can be kept of the same order as standard attention mechanism is promising.

# Weaknesses
1. The paper does not define explicitly some of its notations and concepts when first used, making it difficult to read:
Q in (2) is not defined and the notion of "anchor state" mentioned the line below is not defined explicitly until later to be $z_i^C$, but this is a point in the observation  space rather than in the state space
With $z_i^C = Cx(t_i)$, $z_i$ as defined in p.3, c.2, l.128 is the same as the one in (1), but the noise terms are written differently and with a different covariance matrix without explanations.
Shortly after, we have to guess that $V_{ij}^C$ is the same as V^C(\Delta t_{ij})
In (5), we have to guess what $\lambda_{Q,k}$ and $\lambda_k$ are
etc
Although the appendix does a better job at presenting the results, the main text should be self-contained.
2. The weighting by $w_{ij}$ seems to be heuristic, and no motivation for the particular forms considered is given beyond obtaining "data-dependent precisions". Once again, this is discussed more in the appendix, but some motivation should be given in the main text.
3. The proposed method does slightly better than ALiBi for $L \in \{ 512, 1024 \}$ but quite a bit worse for $L \in \{ 2048, 4096 \}$ in both performance tables. If ALiBi is indeed a special case of the proposed method, there should be a way to at least match the performance.

---

> ### Author Rebuttal · Authors · 2026-03-30
>
> We thank the reviewer for their careful reading and constructive feedback.  We have revised the Methods section to explicitly define all variables at first use (e.g., $Q = GG^\top$) and clarify that the target of estimation is the latent state expressed in observation space, $\boldsymbol{z}_i^C = \boldsymbol{C} \boldsymbol{x}(t_i)$. We remove the "anchor state'' terminology and instead refer directly to this latent state to avoid ambiguity.
>
> We will also clarify the distinction between the two measurement noise terms. The term $v(t_i)$ models measurement noise on transported past observations (key-side), while $\epsilon_\Gamma$ models irreducible uncertainty in the query observation. These are assigned separate covariances $R$ and $R_\Gamma$ because they play structurally different roles: $R$ is propagated through the dynamics, while $R_{\Gamma}$ is not. This follows the standard treatment of two-sided observation noise in state estimation.
>
> The weighting is not heuristic. The base estimator is the exact maximum likelihood solution under a Gaussian model with a conditional independence approximation. The additional weighting $w_{ij}$ arises from replacing the Gaussian likelihood with a heavy-tailed likelihood (Student-$t$ in the isotropic case), which is standard in M-estimation for robustness. We will clarify this directly in the main text. The two commonly used forms of $w_{ij}$ (exponential and power-law) correspond to Gaussian and Student-$t$ likelihoods, respectively. In particular, the exponential form recovers standard softmax attention under a Gaussian likelihood, while the power-law form arises from a heavy-tailed Student-$t$ model. This connection was not sufficiently emphasized in the original draft and will be made explicit in the main text.
>
> The reviewer correctly notes that ALiBi achieves lower perplexity at L=2048 and L=4096. We want to clarify that this reflects a principled trade-off rather than a failure of the method. ALiBi enforces strict locality by suppressing all long-range interactions via a linear distance penalty. SC-RFA instead integrates distant context with attenuated but nonzero precision, governed by the learned decay parameter b, resulting in improved performance both within the training window and at intermediate context lengths. Table 2 shows that this trade-off is continuously controlled. We performed the primary ablations on b=0.05 because it obtained the best performance inside the training window, while still performing well at longer distances, outperforming ALiBi at up to L=1024.
>
> M2.5 ablates complex rotations from RFA. This is RFA in the ALiBi regime. The perplexity score on Wikitext 103 was 28.58 vs ALiBi’s 28.59 within the context window (effectively equal), and at 4096, M2.5 achieved 26.83 PPL vs ALiBi’s 26.30. ALiBi’s narrow lead at very long contexts appears to be due to its more aggressive linear bias, whereas RFA uses a logarithmic bias. We will discuss this in the text.
>
> The assumptions used by RFA are not assumptions on the generative model that actually produced the data. Rather, these are assumptions on the form of the propagation used when comparing keys to queries. A linear SDE is used for the same reason it is used in the Kalman Filter, which is that it allows for closed-form propagation of the mean and covariance. Closed-form propagation is also the reason a linear model is used in RoPE, though the latter lacks a notion of uncertainty. Furthermore, since RFA is a strict generalization of dot product attention and RoPE, its assumptions are at least as general as these other attention mechanisms. We refer the reviewer to the reply to Reviewer Pzpd for a more detailed discussion.
>
> We agree that the conclusion should discuss the limitations of RFA at greater length and we will revise the text accordingly. In particular, we will discuss the assumptions at greater length, and the prospects for relaxing any of them while maintaining tractability as a subject for future work.

---

> > ### Author Rebuttal · Reviewer_3G3G · 2026-04-02
> >
> > The rebuttal provided by the authors effectively addressed my main concerns.

---

### Decision · Program_Chairs · 2026-04-30

**Decision:**

Accept (spotlight)

**Comment:**

This paper introduces Robust Filter Attention (RFA), which reformulates self-attention as parallel robust filtering under a latent SDE prior, where analytically propagated uncertainty defines time-dependent precision over attention weights while preserving standard attention complexity.

The reviewers were broadly enthusiastic. There was broad consensus that this paper introduces a creative idea and executes it well. The scope of the experiments could be a bit broader, but overall this is a strong paper that is likely to be widely read among folks studying attention mechanisms.